# INTERMASK: 3D HUMAN INTERACTION GENERATION VIA COLLABORATIVE MASKED MODELING

**Muhammad Gohar Javed**[1], **Chuan Guo**[2], **Li Cheng**[1], **Xingyu Li**[1]

[1]University of Alberta  [2]Snap Inc.

{javed4,lcheng5,xingyu}@ualberta.ca, cguo2@snapchat.com

## ABSTRACT

Generating realistic 3D human-human interactions from textual descriptions remains a challenging task. Existing approaches, typically based on diffusion models, often produce results lacking realism and fidelity. In this work, we introduce *InterMask*, a novel framework for generating human interactions using collaborative masked modeling in discrete space. InterMask first employs a VQ-VAE to transform each motion sequence into a 2D discrete motion token map. Unlike traditional 1D VQ token maps, it better preserves fine-grained spatio-temporal details and promotes *spatial awareness* within each token. Building on this representation, InterMask utilizes a generative masked modeling framework to collaboratively model the tokens of two interacting individuals. This is achieved by employing a transformer architecture specifically designed to capture complex spatio-temporal inter-dependencies. During training, it randomly masks the motion tokens of both individuals and learns to predict them. For inference, starting from fully masked sequences, it progressively fills in the tokens for both individuals. With its enhanced motion representation, dedicated architecture, and effective learning strategy, InterMask achieves state-of-the-art results, producing high-fidelity and diverse human interactions. It outperforms previous methods, achieving an FID of 5.154 (vs 5.535 of in2IN) on the InterHuman dataset and 0.399 (vs 5.207 of InterGen) on the InterX dataset. Additionally, InterMask seamlessly supports reaction generation without the need for model redesign or fine-tuning.

🌐 gohar-malik.github.io/intermask

## 1 INTRODUCTION

3D human interaction generation is a fundamental task in computer vision and graphics, with applications in animation, virtual reality, robotics, and sports. Although significant progress has been made in generating single-person motion sequences from text descriptions (Tevet et al., 2023; Guo et al., 2024; 2022a; Pinyoanuntapong et al., 2024; Zhang et al., 2023; Petrovich et al., 2022), generating two-person interactions remains a significant challenge. In two-person interactions, each individual's motion depends not only on their own state but also on the state of their interacting partner. Additionally, precise spatial positioning and orientation becomes crucial, specially in close-contact cases. These factors add to the complexity and nuances of human interactions resulting in a higher degree of spatial and temporal interdependence compared to single-person motion.

Existing works on human interaction generation commonly rely on diffusion models. Com-MDM (Shafir et al., 2024) bridges two pretrained single-person motion diffusion models using a small neural layer with limited interaction data. In contrast, InterGen (Liang et al., 2024) proposes a specifically tailored diffusion model for two-person interactions, which uses cooperative transformers to model each individual's motion conditioned on the latent features of their partner's motion. in2IN (Ruiz-Ponce et al., 2024) then extends it by conditioning the diffusion model on LLM-generated individual descriptions in addition to the overall interaction descriptions, and leveraging motion priors learned from single-person motion datasets like HumanML3D (Guo et al., 2022a). MoMat-MoGen Cai et al. (2024) retrieves motions from supplementary datasets, based on textual inputs, and refines them for interaction quality. Some very recent efforts (Shan et al., 2024; Fan et al., 2024) extend these methods to more than two individuals. Despite these efforts, the generated two-person interactions still fall short of achieving satisfactory realism and fidelity.

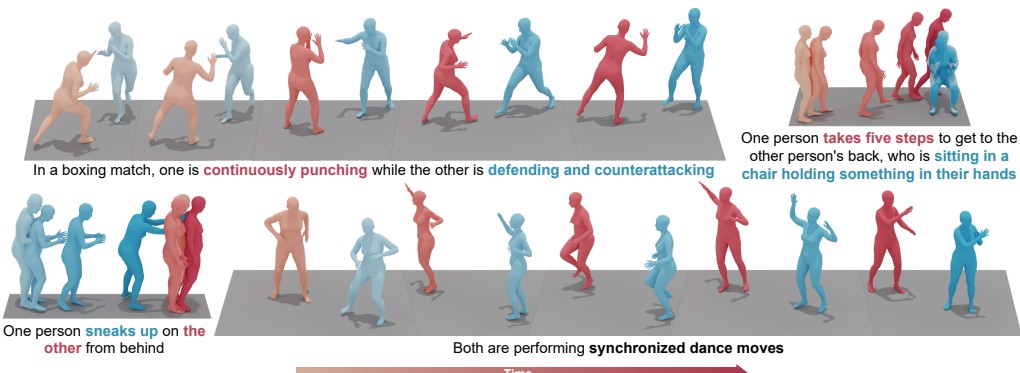

Figure 1: InterMask generates high fidelity text-conditioned 3D human interactions, with accurate spatial and temporal coordination, including *synchronized dance moves*, *realistic reaction timings in boxing*, and *correct proximity*, while maintaining high-quality poses.

In this work, we present *InterMask*, a novel framework based on generative masked modeling in discrete space (Chang et al., 2022). InterMask consists of two stages. First, a shared VQ-VAE is trained to transform each individual's motion into a 2D discrete token map using a learned codebook $\mathcal{C}$ (section 3.1). Unlike previous motion VQ-VAEs (Guo et al., 2022b; Zhang et al., 2023; Jiang et al., 2023; Guo et al., 2024; Pinyoanuntapong et al., 2024), which represent motion as a temporal sequence of 1D pose vectors and use 1D convolutions to capture only temporal context in each token, we preserve both temporal and spatial dimensions. Each motion is represented as $\mathbb{R}^{N \times J \times d}$, where $N$ is the number of poses, $J$ is the number of joints, and $d$ is the joint feature dimension. Using 2D convolutions, we generate a 2D token map, which captures body part context over short time ranges in each token, allowing for better spatial awareness. This is crucial for interactions, where relative body part positioning (e.g., hands) of two individuals holds significant importance.

In the second stage, a specialized Inter-M Transformer (section 3.2) is trained to collaboratively model the tokens of both individuals using a generative masked modeling framework. During training, a random proportion of tokens from either both individuals or only one individual is masked and predicted. The masking proportion follows a scheduling function to ensure its tractability during inference. The inference process begins with all tokens masked, which are progressively generated over a pre-defined number of iterations. At each iteration, the model predicts all masked tokens, but only the most confident predictions are retained, while the rest are re-masked and re-predicted in subsequent iterations. In addition to the standard self attention module, Inter-M Transformer features a shared spatio-temporal attention module to capture fine-grained dependencies within each individual's motion, and a cross-attention module to capture dependencies between the two individuals' motion. This design effectively models the dynamics of spatial and temporal relationships among and between the two individuals, allowing enhanced interaction generation capabilities.

The main contributions of this work are as follows. First, we introduce InterMask, a novel masked generative framework for human-human interaction generation from textual descriptions. Inter-Mask utilizes a 2D VQ encoding that transforms motion sequences into discrete 2D token maps and features a dedicated transformer trained with a generative masked modeling framework which effectively captures both intra- and inter-person spatial and temporal dependencies. Second, Inter-Mask achieves state-of-the-art results in generating high-fidelity, text-conditioned human interactions. Empirically, it achieves an FID of 5.15 (vs. 5.54 of Ruiz-Ponce et al. (2024)) on InterHuman and 0.399 (vs. 5.207 of Liang et al. (2024)) on InterX. Third, InterMask seamlessly supports tasks such as reaction generation without requiring any task-specific fine-tuning or architectural changes.

## 2 RELATED WORK

**Quantized Motion Representation** Quantized latent representations transform continuous data into discrete tokens, which has been widely explored in human motion generation. Deep Motion Signatures (Aristidou et al., 2018) adopted the use of contrastive learning to create discrete motif words,

while TM2T (Guo et al., 2022b) was one of the first to apply VQ-VAE (Van Den Oord et al., 2017) to human motion data, producing discrete motion tokens. T2M-GPT (Zhang et al., 2023) used Exponential Moving Average (EMA) and code reset techniques to improve the VQ-VAE, later adopted in PoseGPT (Lucas et al., 2022) and MotionGPT (Jiang et al., 2023; Zhang et al., 2024b). MoMask (Guo et al., 2024) reduced quantization errors via residual quantization, while MMM (Pinyoanuntapong et al., 2024) utilized a large codebook with the factorized codes technique from Yu et al. (2022). Nevertheless, these works often ignore the spatial dimension of motions, where each token encapsulates the whole information of poses in a short range. This limits the locality of each token, which is important in applications like interactions and motion editing.

**Generative Human Motion Modeling** Synthesizing single-person motion has gained interest, driven by the availability of large motion capture datasets and advancements in generative modeling techniques like Diffusions (Tevet et al., 2023; Kim et al., 2023a; Zhang et al., 2024a; Tseng et al., 2023; Chen et al., 2023; Kong et al., 2023; Lou et al., 2023) and Autoregressive models (Guo et al., 2022b; Zhang et al., 2023; Jiang et al., 2023; Zhang et al., 2024b; Gong et al., 2023; Lucas et al., 2022; Gong et al., 2023). While these models produce high-quality motions, they require numerous sampling steps during inference. Techniques like the DDIM sampler (Song et al., 2021) aim to address this for diffusion models but are still evolving. Additionally, autoregressive models may have limited expressivity since the model only considers past tokens to generate the next one. MoMask (Guo et al., 2024) and MMM (Pinyoanuntapong et al., 2024) attempted to address both these issues by using a masked generative bidirectional transformer to predict all masked tokens simultaneously. These works inspired the base structure of our InterMask.

**Generative Human-Human Interaction Modeling** Most research on human interaction modeling has targeted two-person interactions, primarily addressing two major tasks: 1) reaction generation, where the reactor's motion is generated based on the actor's motion, and 2) interaction generation, where both individuals' motions are generated simultaneously. While reaction generation has seen increasing interest Chopin et al., 2023; Liu et al., 2023; Xu et al., 2024b; Ghosh et al., 2024; Ren et al., 2024; Liu et al., 2024, interaction generation is less explored. ComMDM (Shafir et al., 2024) trained a small neural network to bridge two single-person motion diffusion model (MDM (Tevet et al., 2023)) copies on a limited interaction dataset. On the other hand, RIG (Tanaka & Fujiwara, 2023) and InterGen (Liang et al., 2024) introduced interaction diffusion models that simultaneously denoise both individuals' motions, conditioned on their partner's latent representation. Subsequent works, like Cai et al. (2024) and Ruiz-Ponce et al. (2024) extended interaction diffusion models by using additional single-person annotations and supplementary motion datasets, whereas Shan et al. (2024) and Fan et al. (2024) extended them to handle more than two individuals. While these methods achieve impressive results, there remains significant room for improvement in the quality and realism of two-person interactions. In this paper, we address this by explicitly modeling the spatio-temporal dependencies between interacting individuals' using a masked generative framework.

## 3 METHODOLOGY

Our objective is to generate two-person interaction $\{\mathbf{m}_p\}_{p \in \{a,b\}}$ given a textual description, where $\mathbf{m}_p \in \mathbb{R}^{N \times J \times d}$ represents the motion sequences of one individual (either $a$ or $b$), consisting of $N$ poses, each with $J$ joints and $d$-dimensional joint features. As shown in Figure 2, our methodology consists of two stages. First, we learn a discrete representation of individual motions using a VQ-VAE (section 3.1), which maps motion sequences to a 2-dimensional token map $\{t_p\}_{p \in \{a,b\}} \in \{0, 1, \cdots, |\mathcal{C}| - 1\}^{n \times j}$, where $n$ and $j$ are down-sampled temporal and spatial dimensions, and $|\mathcal{C}|$ is the number of codes in the codebook. Then we learn an interaction masked generative transformer (Inter-M Transformer), (section 3.2) to collaboratively model the discrete tokens of both individuals.

### 3.1 2D DISCRETE MOTION TOKEN MAP

We start by learning a discrete latent representation space for individual motions using a VQ-VAE. We tokenize individual motions instead of the entire interaction because the complexity of modeling interactions is significantly higher. This allows both individuals to share the same discrete token space, simplifying the representation and improving expressivity, as empirically shown in Table 2. Additionally, this provides greater flexibility for complementary tasks such as reaction generation.

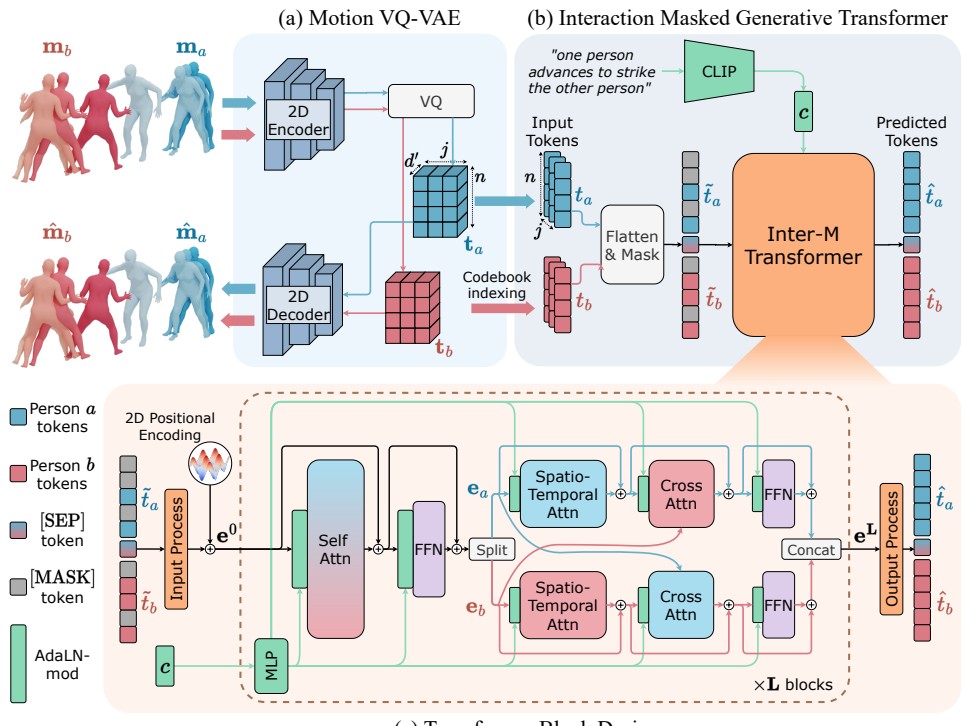

Figure 2: Overview of **InterMask**. (a) Individual motions are quantized through vector quantization (VQ) to obtain 2D tokens $\{t_a, t_b\}$ for each. (b) Motion tokens from both individuals are flattened, concatenated, masked and predicted collaboratively by the Inter-M Transformer. (c) Each block in Inter-M Transformer consists of Self, Spatio-Temporal and Cross Attention modules to learn complex spatio-temporal dependencies within and between both interacting individuals.

We create a 2D token map with both spatial and temporal dimensions by representing motion as $\mathbf{m}_p \in \mathbb{R}^{N \times J \times d}$ and using 2D convolutions. As illustrated in Figure 2(a), our VQ-VAE downsamples the input sequence $\mathbf{m}_p$ into a latent representation $\tilde{\mathbf{t}}_\mathbf{p} \in \mathbb{R}^{n \times j \times d'}$, with downsampling ratios $n/N$ and $j/J$. Each $d'$-dimensional vector is then quantized by replacing it with the nearest entry from a learnable codebook $\mathcal{C} = \{\mathbf{c}_k\}_{k=0}^{|\mathcal{C}|-1} \subset \mathbb{R}^{d'}$, producing a quantized sequence $\mathbf{t}_p = \mathrm{Q}(\tilde{\mathbf{t}}_p) \in \mathbb{R}^{n \times j \times d'}$. This is then decoded to reconstruct the motion $\hat{\mathbf{m}}_\mathbf{p} = \mathrm{D}(\mathbf{t}_\mathbf{p})$. After training, each vector in $\mathbf{t}_p$ can be replaced by its respective index $k$ in $\mathcal{C}$ to obtain the 2d discrete representation of motion, namely *motion tokens* $\{t_p\}_{p \in \{a,b\}} \in \{0, 1, \cdots, |\mathcal{C}| - 1\}^{n \times j}$, which is modeled in our Inter-M Transformer. More details are provided in appendix B.

### 3.1.1 TRAINING OBJECTIVE

Our primary VQ-VAE training objective consists of a motion reconstruction loss and the commitment loss (Van Den Oord et al., 2017):

$$\mathcal{L}_{vq} = \|\mathbf{m}_p - \hat{\mathbf{m}}_p\|_1 + \beta \|\tilde{\mathbf{t}}_p - \mathrm{sg}(\mathbf{t}_p)\|_2^2, \tag{1}$$

where $\mathrm{sg}(\cdot)$ denotes the stop-gradient operation, and $\beta$ a weighting factor. We use Exponential Moving Average (EMA) and codebook reset to update $\mathcal{C}$, following Zhang et al. (2023).

We also incorporate geometric losses from Liang et al. (2024) to impose additional constraints, including the joint velocity loss $\mathcal{L}_{vel}$, the foot contact loss $\mathcal{L}_{fc}$, and the bone length loss $\mathcal{L}_{bl}$, which are explained in appendix C. The overall loss function $\mathcal{L}_{vqvae}$ is a weighted sum of these losses:

$$\mathcal{L}_{vqvae} = \mathcal{L}_{vq} + \lambda_{vel}\mathcal{L}_{vel} + \lambda_{fc}\mathcal{L}_{fc} + \lambda_{bl}\mathcal{L}_{bl} \tag{2}$$

## 3.2 Interaction Masked Generative Transformer

Next, the motion tokens for both individuals $\{t_a, t_b\}$ are collaboratively modeled by our Inter-M Transformer, capturing complex spatial and temporal dependencies within and between them. Importantly, there is no order to the actions of interacting individuals in this task. Therefore we make sure our data sampling, transformer architecture and embedding design are all permutation invariant.

### 3.2.1 Token Masking and Embedding

As shown in Figure 2(b), we first flatten and concatenate $t_a$ and $t_b$, separated by a special [SEP] token, to form a consolidated sequence $t \in \mathbb{Z}^{2nj+1}$. A subset of these tokens is then masked by replacing with a learnable special-purpose token [MASK] following a two-stage scheme. In the first stage, we apply *random masking* with probability $p_r$ or *interaction masking* with $1 - p_r$. In random masking, both $t_a$ and $t_b$ are randomly masked with ratio $\gamma(\tau_i)$, controlled by a cosine scheduling function (Chang et al., 2022), as shown in eq. (3). In interaction masking, one of $\{t_a, t_b\}$ is kept fully unmasked, encouraging the model not to rely on the tokens from the same individual and learn inter-individual dependencies.

$$\gamma(\tau_i) = \cos\left(\frac{\pi \tau_i}{2}\right) \in [0, 1]$$
$$\tau_i \sim \mathcal{U}(0, 1) \tag{3}$$

In the second stage, we employ *step unroll masking*, adopted from Kim et al. (2023b), where we remask a portion ($\gamma(\tau_{i+1})$) of the predicted tokens with the lowest confidence scores and predict them again. This refines the predictions iteratively, aligning with the inference process, where tokens are progressively revealed. More details of the masking strategy are provided in appendix D. The masked tokens $\tilde{t}$ are then embedded using the *input process* module, consisting of a learnable embedding layer and a linear transformation. Then, 2D positional encodings, from Wang & Liu (2021), are added to impose spatio-temporal structure on the embeddings. The updated embeddings, $\mathbf{e}^0 \in \mathbb{R}^{(2nj+1) \times \tilde{d}}$, where $\tilde{d}$ is the transformer embedding dimension, are then passed through **L** blocks of the Inter-M transformer.

### 3.2.2 Transformer block design

As illustrated in Figure 2(c), each Inter-M block starts with a self-attention module (Vaswani et al., 2017) to model long-range, global dependencies within and between individuals. Then we introduce a novel shared spatio-temporal attention layer to model spatial and temporal dependencies within each individual's motion tokens. Finally, we employ a shared cross attention layer to model inter-individual dependencies. This design enables rich and dynamic modeling of spatio-temporal interactions, ensuring effective learning of both intra- and inter-individual relationships.

**Self Attention** Let $\mathbf{e}^{l-1} \in \mathbb{R}^{(2nj+1) \times \tilde{d}}$ represent the input token embeddings to block $l$, where $l \in \{1, 2, \ldots \mathbf{L}\}$. The block architecture begins with a self-attention module which computes attention scores for all tokens in the sequence using the standard scaled dot-product attention mechanism:

$$\text{Attn}(\mathbf{Q}, \mathbf{K}, \mathbf{V}) = \text{softmax}(\mathbf{Q}\mathbf{K}^\top / \sqrt{\tilde{d}})\mathbf{V} \tag{4}$$

where the query, key, and value matrices, $\mathbf{Q}$, $\mathbf{K}$, and $\mathbf{V}$, are linear projections of $\mathbf{e}_{l-1}$. The self attention module is followed by a feedforward network (FFN).

**Spatio-Temporal Attention** Next, we split the embeddings of the two individuals and pass them through a shared spatio-temporal attention module. This module consists of two separate attention mechanisms: spatial attention and temporal attention. In the spatial attention module, each token attends only to other spatial tokens within the same temporal instance; and in the temporal attention module, each token attends only to tokens across the temporal dimension at the same spatial location. Let $\mathbf{e}_p^{l-1}$ represent the updated embeddings of individual $p \in \{a, b\}$ in block $l$ after the self attention module. We drop the $l - 1$ superscript to simplify the notation. Then $\mathbf{e}_p(i_n) \in \mathbb{R}^{j \times \tilde{d}}$ represent the spatial tokens of individual $p$ at temporal instance $i_n$, and $\mathbf{e}_p(i_j) \in \mathbb{R}^{n \times \tilde{d}}$ represent the temporal tokens of individual $p$ at spatial location $i_j$. The spatial and temporal attention is defined as:

$$\mathbf{e}_p'(i_n) = \text{Attn}(\mathbf{Q}_j, \mathbf{K}_j, \mathbf{V}_j) \quad \mathbf{e}_p''(i_j) = \text{Attn}(\mathbf{Q}_n, \mathbf{K}_n, \mathbf{V}_n) \tag{5}$$

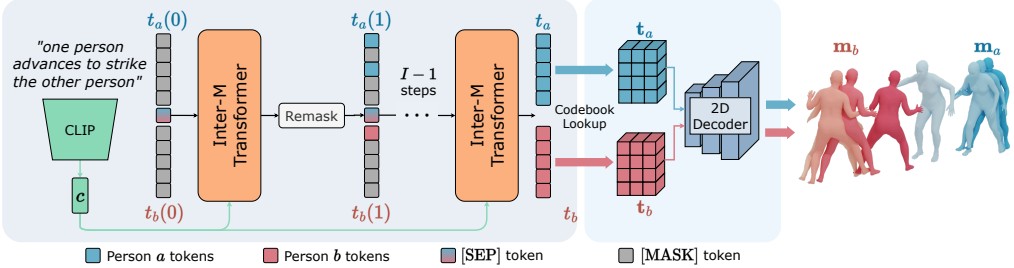

Figure 3: Inference process. Starting from completely masked token sequences of both individuals $\{t_a(0), t_b(0)\}$, the Inter-M transformer generates all tokens in $I$ iterations. Next, the tokens are dequantized and decoded to generate motion sequences $\{\mathbf{m}_a, \mathbf{m}_b\}$ using the VQ-VAE decoder.

where $\mathbf{Q}_j, \mathbf{K}_j, \mathbf{V}_j$, and $\mathbf{Q}_n, \mathbf{K}_n, \mathbf{V}_n$ are the spatial queries, keys and values and temporal queries, keys, and values, obtained from $\mathbf{e}_p(i_n)$ and $\mathbf{e}_p(i_j)$ respectively. After updating the embeddings with spatial and temporal attention, the outputs of these two modules are added together for each token:

$$\mathbf{e}_p = \mathbf{e}_p'(i_n) + \mathbf{e}_p''(i_j) \quad \forall \quad 0 < i_n < n, 0 < i_j < j \tag{6}$$

**Cross Attention** Next, we apply a shared cross attention module to model dependencies between the two individuals. Each token of individual $a$ attends to all tokens of individual $b$ and vice versa:

$$\mathbf{e}_a' = \mathrm{Attn}(\mathbf{Q}_a, \mathbf{K}_b, \mathbf{V}_b) \quad \mathbf{e}_b' = \mathrm{Attn}(\mathbf{Q}_b, \mathbf{K}_a, \mathbf{V}_a) \tag{7}$$

where $\mathbf{Q}_a, \mathbf{K}_a, \mathbf{V}_a$, and $\mathbf{Q}_b, \mathbf{K}_b, \mathbf{V}_b$ are the queries, keys and values obtained from embeddings $\mathbf{e}_a, \mathbf{e}_b$ of each individual. Finally, each individual's embeddings are passed through another FFN and concatenated to form the sequence $\mathbf{e}^l \in \mathbb{R}^{(2nj+1) \times \tilde{d}}$, which is passed to the next block.

**Text Conditioning** We condition our transformer blocks on the input text by replacing the standard layer normalization in each attention and FFN module with modulated adaptive layer normalization (AdaLN-mod) as proposed by Peebles & Xie (2023). In AdaLN-mod, the dimension-wise scale and shift parameters are regressed using a multi-layer perceptron (MLP), from the conditional vector $c$, obtained from a frozen, pre-trained CLIP network (Radford et al., 2021). This modulation allows the model to dynamically adapt its normalization based on the content of $c$. Additionally, we initialize each residual connection in the transformer block as an identity function by regressing another scale parameter $\alpha$ from $c$, which is initialized to output a zero vector. This ensures that the network can adapt the modulation during learning without imposing strong biases at initialization.

### 3.2.3 OUTPUT PROCESS

After the final block, we use a standard linear projector with AdaLN-mod to map the output $\mathbf{e}^{\mathbf{L}}$ to the indices of codebook $\mathcal{C}$, $\mathbf{e}_{out} \in \mathbb{R}^{2nj \times |\mathcal{C}|}$, where $|\mathcal{C}|$ represents the size of $\mathcal{C}$. As defined in eq. (8), the transformer's training objective minimizes the negative log-likelihood of the masked tokens, computed as the cross-entropy between the one-hot encoded ground truth and model predictions.

$$\mathcal{L}_{mask} = \sum_{\tilde{t}_k = [\mathrm{MASK}]} -\log p_\theta(t_k | \tilde{t}, c). \tag{8}$$

### 3.3 INFERENCE

As illustrated in Figure 3, the inference process consists of two stages. Starting with a fully masked sequence $t(0)$, the Inter-M transformer generates tokens for both individuals over $I$ iterations. At each iteration $i$, the transformer predicts token probabilities at masked locations, then samples tokens and remasks those with the lowest $\lceil \gamma \left( \frac{i}{I} \right) \cdot 2nj \rceil$ confidence scores, repeating until $i$ reaches $I$. The final token sequence is decoded back into motion using the VQ-VAE decoder. A cosine schedule $\gamma \left( \frac{i}{I} \right)$ controls the number of retained tokens, increasing as $i$ progresses. Classifier-free guidance (Ho & Salimans, 2021) is also applied for refinement, following Chang et al. (2022).

| Dataset | Method | R Precision↑ | | | FID↓ | MM Dist↓ | Diversity→ | MModality↑ |
|---|---|---|---|---|---|---|---|---|
| | | Top 1 | Top 2 | Top 3 | | | | |
| Inter Human | Ground Truth | $0.452^{\pm.008}$ | $0.610^{\pm.009}$ | $0.701^{\pm.008}$ | $0.273^{\pm.007}$ | $3.755^{\pm.008}$ | $7.948^{\pm.064}$ | - |
| | T2M (Guo et al., 2022a) | $0.238^{\pm.012}$ | $0.325^{\pm.010}$ | $0.464^{\pm.014}$ | $13.769^{\pm.072}$ | $5.731^{\pm.013}$ | $7.046^{\pm.022}$ | $1.387^{\pm.076}$ |
| | MDM (Tevet et al., 2023) | $0.153^{\pm.012}$ | $0.260^{\pm.009}$ | $0.339^{\pm.012}$ | $9.167^{\pm.056}$ | $7.125^{\pm.018}$ | $7.602^{\pm.045}$ | $\mathbf{2.350}^{\pm.080}$ |
| | ComMDM (Shafir et al., 2024) | $0.223^{\pm.009}$ | $0.334^{\pm.008}$ | $0.466^{\pm.010}$ | $7.069^{\pm.054}$ | $6.212^{\pm.021}$ | $7.244^{\pm.038}$ | $1.822^{\pm.052}$ |
| | InterGen (Liang et al., 2024) | $0.371^{\pm.010}$ | $0.515^{\pm.012}$ | $0.624^{\pm.010}$ | $5.918^{\pm.079}$ | $5.108^{\pm.014}$ | $7.387^{\pm.029}$ | $\underline{2.141}^{\pm.063}$ |
| | MoMat-MoGen (Cai et al., 2024) | $\mathbf{0.449}^{\pm.004}$ | $\underline{0.591}^{\pm.003}$ | $\underline{0.666}^{\pm.004}$ | $5.674^{\pm.085}$ | $\mathbf{3.790}^{\pm.001}$ | $8.021^{\pm.35}$ | $1.295^{\pm.023}$ |
| | in2IN (Ruiz-Ponce et al., 2024) | $\underline{0.425}^{\pm.008}$ | $0.576^{\pm.008}$ | $0.662^{\pm.009}$ | $\underline{5.535}^{\pm.120}$ | $\underline{3.803}^{\pm.002}$ | $\underline{7.953}^{\pm.047}$ | $1.215^{\pm.023}$ |
| | **InterMask** | $\mathbf{0.449}^{\pm004}$ | $\mathbf{0.599}^{\pm.005}$ | $\mathbf{0.683}^{\pm004}$ | $\mathbf{5.154}^{\pm.061}$ | $\mathbf{3.790}^{\pm.002}$ | $\mathbf{7.944}^{\pm.033}$ | $1.737^{\pm.020}$ |
| InterX | Ground Truth | $0.429^{\pm.004}$ | $0.626^{\pm.003}$ | $0.736^{\pm.003}$ | $0.002^{\pm.0002}$ | $3.536^{\pm.013}$ | $9.734^{\pm.078}$ | - |
| | T2M (Guo et al., 2022a) | $0.184^{\pm.010}$ | $0.298^{\pm.006}$ | $0.396^{\pm.005}$ | $5.481^{\pm.382}$ | $9.576^{\pm.006}$ | $2.771^{\pm.151}$ | $2.761^{\pm.042}$ |
| | MDM (Tevet et al., 2023) | $0.203^{\pm.009}$ | $0.329^{\pm.007}$ | $0.426^{\pm.005}$ | $23.701^{\pm.057}$ | $\underline{9.548}^{\pm.014}$ | $5.856^{\pm.077}$ | $\underline{3.490}^{\pm.061}$ |
| | ComMDM (Shafir et al., 2024) | $0.090^{\pm.002}$ | $0.165^{\pm.004}$ | $0.236^{\pm.004}$ | $29.266^{\pm.067}$ | $6.870^{\pm.017}$ | $4.734^{\pm.067}$ | $0.771^{\pm.053}$ |
| | InterGen (Liang et al., 2024) | $\underline{0.207}^{\pm.004}$ | $\underline{0.335}^{\pm.005}$ | $\underline{0.429}^{\pm.005}$ | $\underline{5.207}^{\pm.216}$ | $9.580^{\pm.011}$ | $\underline{7.788}^{\pm.208}$ | $\mathbf{3.686}^{\pm.052}$ |
| | **InterMask** | $\mathbf{0.403}^{\pm.005}$ | $\mathbf{0.595}^{\pm.004}$ | $\mathbf{0.705}^{\pm005}$ | $\mathbf{0.399}^{\pm.013}$ | $\mathbf{3.705}^{\pm.017}$ | $\mathbf{9.046}^{\pm.073}$ | $2.261^{\pm.081}$ |

Table 1: **Quantitative evaluation** on the **InterHuman** and **InterX** test sets. $\pm$ indicates a 95% confidence interval and $\rightarrow$ means the closer to ground truth the better. **Bold** face indicates the best result, while underscore refers to the second best.

## 4 EXPERIMENTS

**Datasets** We adopt two datasets to evaluate InterMask for the text-conditioned human interaction generation task: InterHuman (Liang et al., 2024) and InterX (Xu et al., 2024a). The InterHuman dataset contains 7,779 interaction sequences and InterX contains 11,388, each paired with 3 distinct textual annotations. InterHuman follows the AMASS (Mahmood et al., 2019) skeleton representation with 22 joints, including the root joint. Each joint is represented by $\{p_g, v_g, r_{6d}\}$, where $p_g \in \mathbb{R}^3$ is the global position, $v_g \in \mathbb{R}^3$ is the global velocity, and $r_{6d} \in \mathbb{R}^6$ is the local 6D rotation of each joint, rendering $\mathbf{m}_p \in \mathbb{R}^{N \times 22 \times 12}$. InterX follows the SMPL-X (Pavlakos et al., 2019) skeleton representation, comprising 54 body, hands and face joints, accompanied by root orientation and translation. Each joint and root orientation is represented by $\{r_{6d}\}$ and root translation by $\{p_g\}$, to which we incorporate root velocity rendering root $\{p_g, v_g\}$ and $\mathbf{m}_p \in \mathbb{R}^{N \times 56 \times 6}$. We adhere to the respective body representations of both datasets to demonstrate that our framework is compatible with different representation formats and joint counts. Other implementation details including architecture, training and inference hyper-parameters are provided in appendix E.

**Evaluation Metrics** Following Liang et al. (2024), we adopt several feature-space evaluation metrics to assess the performance of our model. These include the Frechet Inception Distance (*FID*), which measures the fidelity of the generated interactions by calculating similarity between the generated and real interactions feature distributions. Additionally we employ *R-precision* and *MMDist*, which evaluate how well the generated interactions align with the corresponding texts. Furthermore, we use *Diversity* to measure the overall variation in generated interactions, and multimodality (*MModality*) to quantify the ability to generate multiple distinct interactions for the same text.

### 4.1 COMPARISON WITH STATE-OF-THE-ART APPROACHES

**Quantitative Comparison** Table 1 shows quantitative comparison of our **InterMask** with previous methods. All evaluations are run 20 times (with the exception of MModality, which is run 5 times), and we report the averaged results along with a 95% confidence interval. InterMask achieves state-of-the-art (SOTA) results on both InterHuman and InterX datasets. It records the lowest FID scores (5.154 on InterHuman and 0.399 on InterX), indicating superior realism and quality of generated interactions, and leads in R-Precision and MMdist, showing excellent text alignment. With different body representations in each dataset, these results highlight the robustness of our method, proving it is not dependent on a specific body pose structure. Furthermore, the larger performance gap on the larger InterX dataset suggests that InterMask scales effectively with more data. While our MModality is slightly lower than some methods, the high R-Precision and low MMdist emphasize that InterMask prioritizes adherence to text over extreme diversity, yet still achieves high Diversity scores, demonstrating its ability to generate a broad range of interactions. Please refer to appendix F for visual demonstration of InterMask's ability to generate diverse results from the same text prompt.

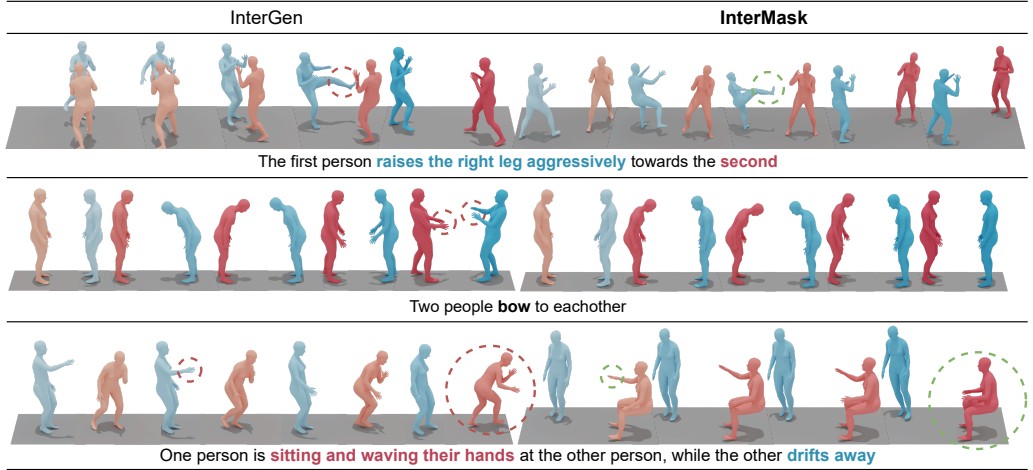

Figure 4: Qualitative comparison between InterMask and InterGen (Liang et al., 2024), highlighting InterMask's superior interaction quality, text adherence and avoidance of implicit biases.

| Input Modality | Token Map | Recon FID↓ | MPJPE ↓ |
|---|---|---|---|
| Individual Motion | 1d | 3.146 | 0.354 |
| Two-Person Interaction | 2d | 1.276 | 0.198 |
| Individual Motion | 2d | **0.970** | **0.129** |

Table 2: Ablation Study results on InterHuman test set to verify key components of the proposed **Motion VQ-VAE**. **Bold** face indicates the best result.

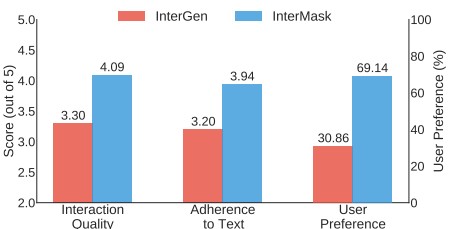

Figure 5: User Study comparing our Inter-Mask and InterGen (Liang et al., 2024).

**Qualitative Comparison** In Figure 4, we provide a qualitative comparison of interaction sequences generated by our InterMask and InterGen Liang et al. (2024), trained on the InterHuman dataset, for the same text descriptions. For the prompt "*The first person raises the right leg aggressively towards the second*", InterGen incorrectly raises both legs, while InterMask accurately raises only the right leg. For "*Two people bow to each other*", InterGen introduces an unnecessary fighting gesture after the bow, indicating overfitting to implicit biases in the training data, whereas InterMask stays true to the simple bow. Lastly, for "*One person is sitting and waving their hands at the other person, while the other drifts away*", InterGen generates a crouching pose instead of sitting and misplaces the waving action, while InterMask faithfully generates the described scene. These examples demonstrate that InterMask produces more higher quality and more reliable interactions compared to InterGen. For more qualitative results please refer to appendix A.

**User Study** User studies are essential in generative AI, complementing metrics like FID by incorporating human perception to evaluate the realism and quality of generated content. To compare our **InterMask** with InterGen (Liang et al., 2024), we conducted a user study with 16 participants, where 30 interaction sequences were generated by both models with identical motion lengths. Each sequence pair was rated by 3 distinct users on interaction quality and adherence to text (scores out of 5), and their preferred interaction sequence among the two. More details are provided in appendix K. As shown in Figure 5, InterMask outperforms InterGen in interaction quality (4.089 vs. 3.296) and text adherence (3.938 vs. 3.198), with 69.14% of users preferring *InterMask* over InterGen.

**Computation and Time Cost** In addition to superior generation fidelity and text adherence, Inter-Mask demonstrates better computational efficiency when compared to InterGen (Liang et al., 2024). InterMask contains only **74** M inference parameters, compared to 182 M for InterGen, resulting in a considerably smaller model footprint. Furthermore, InterMask achieves an average inference time of **0.77** seconds, less than half of InterGen's 1.63 seconds, highlighting its low latency.

## 4.2 ABLATION STUDIES

| Method | R Precision↑ | | | FID↓ | MM Dist↓ | Diversity→ | MModality↑ |
|---|---|---|---|---|---|---|---|
| | Top 1 | Top 2 | Top 3 | | | | |
| Ground Truth | $0.452^{\pm.008}$ | $0.610^{\pm.009}$ | $0.701^{\pm.008}$ | $0.273^{\pm.007}$ | $3.755^{\pm.008}$ | $7.948^{\pm.064}$ | - |
| Alternative Modeling | $0.340^{\pm.002}$ | $0.425^{\pm.009}$ | $0.514^{\pm.006}$ | $7.637^{\pm.072}$ | $3.937^{\pm.003}$ | $8.424^{\pm.022}$ | $\mathbf{2.192}^{\pm.076}$ |
| Collaborative Modeling | | | | | | | |
| w/o Self Attention | $0.393^{\pm.006}$ | $0.527^{\pm.005}$ | $0.596^{\pm007}$ | $5.715^{\pm.067}$ | $3.827^{\pm.003}$ | $7.873^{\pm.034}$ | $1.683^{\pm.076}$ |
| w/o Cross Attention | $0.328^{\pm.008}$ | $0.416^{\pm.009}$ | $0.508^{\pm007}$ | $8.461^{\pm.076}$ | $3.972^{\pm.005}$ | $\mathbf{7.951}^{\pm.085}$ | $1.942^{\pm.076}$ |
| w/o Spatio-Temporal Attention | $0.296^{\pm.004}$ | $0.391^{\pm.007}$ | $0.470^{\pm005}$ | $10.968^{\pm.093}$ | $4.219^{\pm.006}$ | $7.659^{\pm.023}$ | $1.607^{\pm.076}$ |
| w/o Step Unroll and Interaction Masking | $0.402^{\pm.003}$ | $0.548^{\pm.004}$ | $0.633^{\pm002}$ | $5.629^{\pm.053}$ | $3.806^{\pm.002}$ | $7.942^{\pm.096}$ | $1.741^{\pm.076}$ |
| **InterMask** | $\mathbf{0.449}^{\pm.004}$ | $\mathbf{0.599}^{\pm.005}$ | $\mathbf{0.683}^{\pm004}$ | $\mathbf{5.154}^{\pm.061}$ | $\mathbf{3.790}^{\pm.002}$ | $7.944^{\pm.033}$ | $1.737^{\pm.020}$ |

Table 3: Ablation Study results on the InterHuman test set to verify key components of the proposed **Inter-M Transformer**. **Bold** face indicates the best result.

Table 2 presents an ablation study to evaluate key components of our proposed *Motion VQ-VAE*. We investigated two primary factors: the input modality (individual motions vs two-person interactions) and the token map dimensions (1d vs 2d), using Reconstruction FID and MPJPE (Mean Per Joint Position Error). The results demonstrate that using individual motions as input with a 2D token map outperforms both the baseline of individual motions with a 1D token map, and the two-person interaction with a 2D token map. These findings suggest that preserving spatial information through 2D token maps and focusing on individual motions performs accurate motion reconstruction, and creates a more expressive token map with high-fidelity movement details.

Table 3 presents an ablation study to evaluate key components of our *Inter-M Transformer*. We compared collaborative modeling against alternative modeling and examined the impact of various attention mechanisms and masking strategies. *Alternative Modeling* (appendix G) models tokens of one person at a time, while conditioning in on the tokens of the other person, leading to an alternative generation during inference. *Collaborative Modeling* on the other hand, models tokens of both individuals simultaneously. Our full InterMask model with collaborataive modeling outperforms all ablated versions, particularly in R-Precision and FID scores. Notably, removing any of the attention mechanisms or employing alternative modeling severely degraded performance, with spatio-temporal attention being the most crucial one. Additionally, step unroll and interaction masking proved to be further fine-tuning the results. These results highlight the synergistic importance of all components in our transformer architecture for effective interaction modeling. Qualitative results for selected ablation studies are shown in appendix H.

## 4.3 APPLICATION: REACTION GENERATION

One of the key strengths of our InterMask model is its seamless support for the task of reaction generation with or without a text condition, where the motion of one person in an interaction is generated based on the provided reference motion of the other person. Unlike previous diffusion-based models, such as InterGen (Liang et al., 2024), InterMask can perform this task without any task-specific fine-tuning or architectural re-design. This flexibility arises from our use of the masked modeling technique combined with an individual motion tokenizer, which does not require interactions as input. During inference, we simply tokenize the reference motion, leave its tokens unmasked, and progressively generate tokens of the other person. More details are provided in appendix I.

Quantitatively, InterMask achieves an FID of 2.99 and an R-precision of 0.462 on InterHuman for reaction generation. In comparison, InterGen suffers from a performance drop with an FID of 52.89 (vs 5.918 for interaction generation) and an R-precision of 0.194 (vs 0.371) when adopted for reaction generation without fine-tuning, demonstrating its inability to perform reaction generation out-of-the-box. As shown qualitatively in Figure 6, InterMask produces high-quality reaction motions. In cases without text conditioning, when the reference motion involves dancing, the generated motion follows the reference steps and even captures nuanced details like raising the same leg or arm. Similarly, for the physical combat scenario, when the reference motion attacks, the generated motion retreats and vice versa. For text-conditioned samples, InterMask performs equally well, whether the text describes both people's motions or provides separate descriptions for each. The

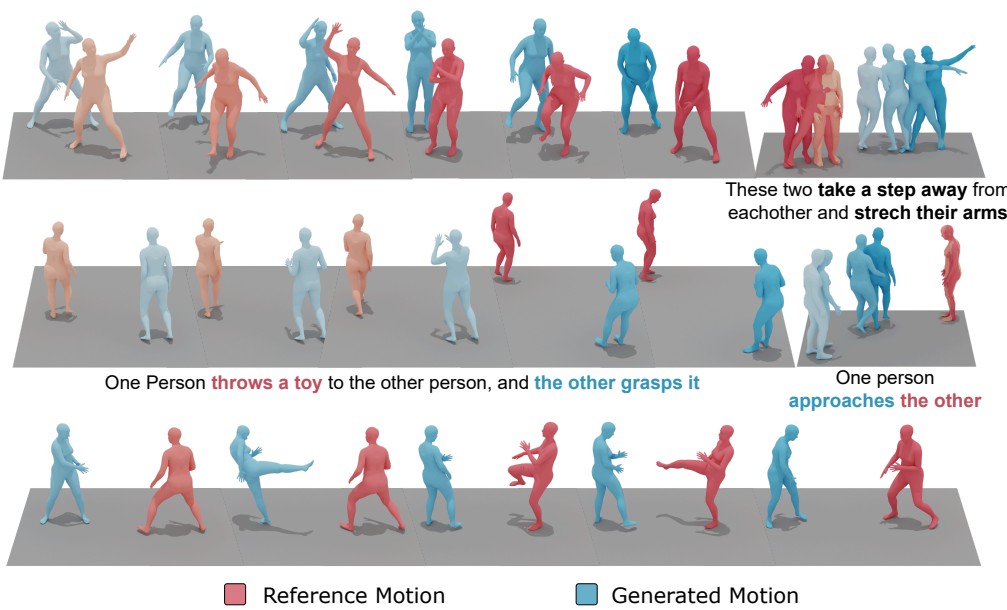

These two **take a step away** from eachother and **strech their arms**

One Person **throws a toy** to the other person, and **the other grasps it**

One person **approaches the other**

🟥 Reference Motion      🟦 Generated Motion

Figure 6: Reaction generation samples with and without text descriptions, including holistic instruction for both individuals and separate instructions for each.

model accurately identifies the required reaction form text and the reference motion and generates the corresponding motion, maintaining high-quality poses and realistic timings throughout.

## 5 CONCLUSION

In this paper, we presented InterMask, a novel framework for generating realistic 3D human interactions through generative masked modeling in the discrete space. InterMask first employs a VQ-VAE to transform individual motion sequences into 2D discrete token maps, preserving both spatial and temporal dimensions. Then, a specialized Inter-M transformer collaboratively models the motion tokens of both individuals, capturing intricate spatial and temporal dependencies within and between them. Quantitative evaluations show that InterMask excels in generating realistic and contextually aligned motions, achieving the lowest FID scores and superior results in R-Precision and MMdist on both the InterHuman and InterX datasets compared to existing models. Additionally, user studies comparing InterMask to InterGen reveal a 69% user preference for InterMask, noting higher-quality and more realistic interactions. The flexibility of our approach is also evident by its ability to perform equally well for different body representations. While there are minor trade-offs in diversity (MModality), InterMask consistently maintains high quality and adherence to text while generating a wide range of realistic interactions.

**Limitations and Future Work** While InterMask generates high-quality human interactions, there are some limitations. First, when visualized as SMPL Loper et al. (2015) meshes, issues such as body penetration between individuals and jerky movements during rapid actions can occur. Second, our model sometimes interprets motions as dances without explicit prompting, likely due to implicit biases in the training dataset. Lastly, our approach is currently optimized for short sequences up to 10 seconds, reflecting the constraints of the dataset. Visual examples are shown in appendix J. Future research directions include implementing techniques for smoother transitions, preventing penetrations, addressing dataset biases, and expanding the model's capacity to handle longer sequences.

**Reproducibility Statement** We have made our best effort to ensure reproducibility, including but not limited to: 1) description of our implementation details in appendix E; 2) detailed graphic illustrations of our model architectures, training mechanisms and inference processes in Figures 2, 3, 7, 8, 10 and 13; and 4) public release of code and model checkpoints (appendix A).

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

## A    SUPPLEMENTARY MATERIAL

In addition to the subsequent appendix sections, we provide the following as supplementary material for this work:

- **Demo Videos**: We provide several demonstration videos showcasing visual interaction results generated using InterMask. These include an animation gallery with generated interactions for everyday actions, dance and combat; nuanced description demonstration to highlight the capability of Intermask to follow specific details in text; diversity demonstration with multiple generated interactions for each text prompt; comparison videos to compare the generated results of InterMask and InterGen (Liang et al., 2024); comparison videos for the ablation study on Inter-M Transformer; an animation gallery to show reaction generation results; some results on more complex prompts; some results showing longer 10 second interactions; and finally some failure cases and some results addressing the concerns on fluidity of the generated motions. The videos are shown in form of a webpage, provided as an html file.

- **Code Implementation**: We provide the open-source code implementation of our method to ensure reproducibility.

$\qquad$ github.com/gohar-malik/intermask

## B    MOTION VQ-VAE

Our VQ-VAE framework, illustrated in Figure 7, constructs a 2D motion token map to represent individual motion sequences in a discrete manner, while retaning both spatial and temporal dimensions. The encoder processes motion sequences represented as $\mathbf{m}_p \in \mathbb{R}^{N \times J \times d}$, where $N$ is the number of poses, $J$ is the number of joints, and $d$ is the joint feature dimension. By employing 2D convolutional layers, the encoder effectively captures spatial and temporal dependencies within the motion data while progressively downsampling both dimensions. The downsampling process is achieved through strided 2D convolutions and ResNet blocks, which also use 2D convolutions along with dropout layers. This results in a latent representation of size $\tilde{\mathbf{t}}_\mathbf{p} \in \mathbb{R}^{n \times j \times d'}$, where $n$ and $j$ are the downsampled temporal and spatial dimensions, and $d'$ is the latent feature dimension.

The latent representation $\tilde{\mathbf{t}}_\mathbf{p}$ is quantized using a learned codebook $\mathcal{C}$ with $|\mathcal{C}|$ entries. Each feature vector $\tilde{t}_i$ in $\tilde{\mathbf{t}}_\mathbf{p}$ is replaced by the index of its nearest codebook entry, using the vector quantization process:

$$\mathbf{q}(\tilde{t}_i) = \arg \min_{c_k \in \mathcal{C}} \|\tilde{t}_i - c_k\|^2, \tag{9}$$

where $c_k$ represents the codebook entries.

The resulting quantized representation is a 2D motion token map $t_p$, where each token encodes local spatio-temporal context. This design enables the preservation of both spatial and temporal dimensions in the motion data, enhancing the model's ability to generate realistic and contextually accurate interactions.

## C    VQ-VAE GEOMETRIC LOSSES

Equation (10) shows the geometric losses used to train our Motion VQ-VAE to impose physical and geometric constraints on the reconstructed motion, in a data-driven way. The velocity loss $\mathcal{L}_{vel}$ encourages the reconstructed motion sequences to obey the velocity of joints in the ground truth sequences and the bone length loss $\mathcal{L}_{bl}$ the distance between adjacent joints. The foot contact loss $\mathcal{L}_f c$ encourages the feet to have zero velocity whenever they are in contact with the ground.

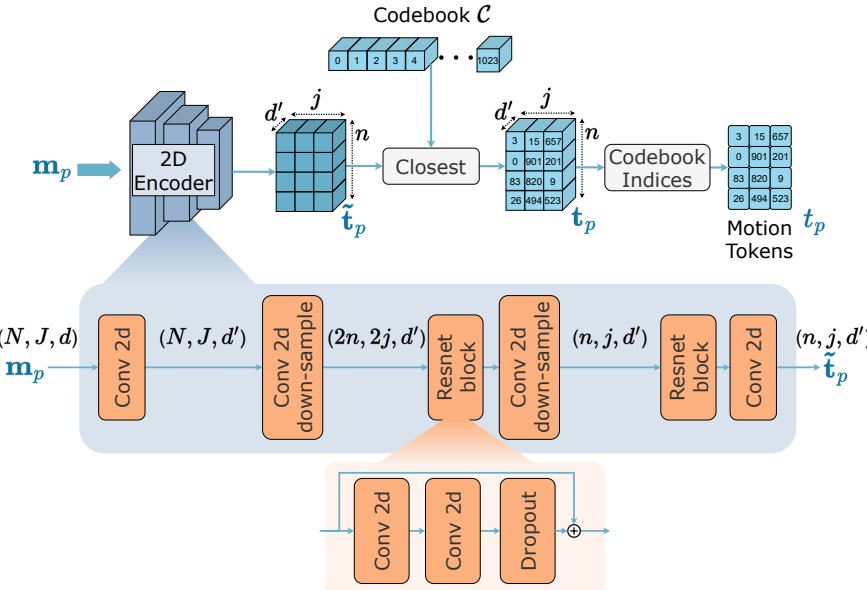

Figure 7: Detailed illustration of the 2d discrete motion token map construction. The 2d encoder, consisting of 2d convolutional layers, downsamples the input motion from $(N, J, d)$ to $(n, j, d')$. The downsampled representation is then quantized by replacing each vector with the index of its closest vector in the learned codebook.

| $p_r$ | Interaction Generation | | Reaction Generation | |
|---|---|---|---|---|
| | FID $\downarrow$ | R Prec (Top1) $\uparrow$ | FID $\downarrow$ | R Prec (Top1) $\uparrow$ |
| 0.7 | 5.214 | 0.447 | **2.850** | **0.476** |
| 0.8 | 5.154 | 0.449 | 2.991 | 0.462 |
| 0.9 | **5.152** | **0.450** | 3.368 | 0.416 |

Table 4: Interaction Generation and Reaction Generation results of different values of $p_r$. **Bold** face indicates the best result, while underscore refers to the second best.

$$\mathcal{L}_{vel} = \frac{1}{N-1} \sum_{i_n=1}^{N} \|(m_{i_n+1} - m_{i_n}) - (\hat{m}_{i_n+1} - \hat{m}_{i_n})\|_1$$

$$\mathcal{L}_{fc} = \frac{1}{N-1} \sum_{i_n=1}^{N} \|(\hat{m}_{i_n+1} - \hat{m}_{i_n}) \cdot f_{i_n}\|_1 \tag{10}$$

$$\mathcal{L}_{bl} = \frac{1}{N-1} \sum_{i_n=1}^{N} \|B(m_{i_n}) - B(\hat{m}_{i_n})\|_1$$

Here, $m_{i_n}$ represents the ground pose, $\hat{m}_{i_n}$ represents the reconstructed pose at time step $i_n$, $N$ represents the total time steps in the sequence, $f_{i_n} \in \{0, 1\}$ represents the binary foot contact label for the heel and toe joints for each pose $m_{i_n}$, and $B(\cdot)$ denotes the bone lengths joining adjacent joints.

## D  TWO-STAGE TOKEN MASKING

Here we provide more details on the masking strategy (section 3.2.1) used in training the Inter-M Transformer. As illustrated in Figure 8, the two-stage masking technique begins with random

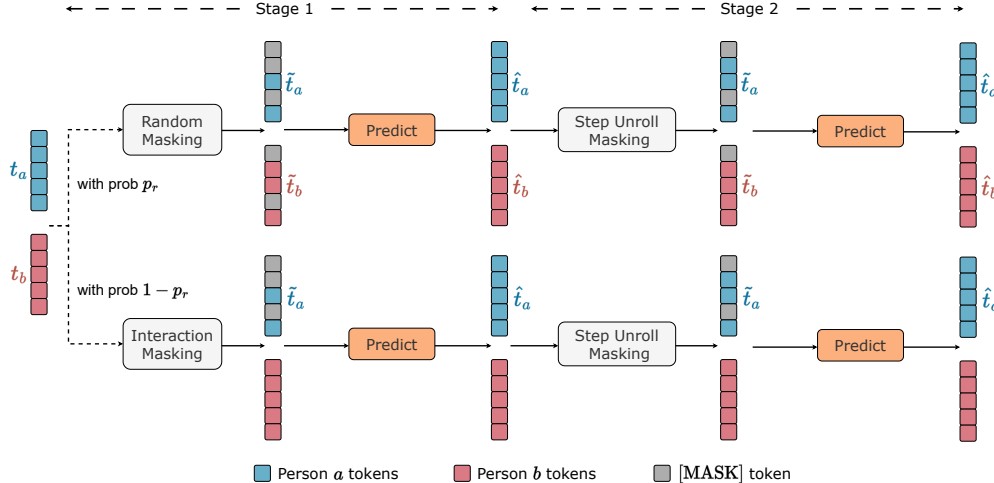

Figure 8: Illustration of the two-stage masking technique used during training of the Inter-M Transformer. For stage 1, we either apply Random Masking with a probability of $p_r$ or Interaction Masking with probability $1 - p_r$. In stage 2, we apply the Step Unroll Masking on the predicted tokens from stage 1.

masking or interaction masking in the first stage. Random masking teaches the model to predict random tokens from both individuals. Whereas Interaction masking, where only one individual's tokens are masked, promotes learning inter-person dependencies critical for coherent interactions and to improve performance in the reaction generation task. The masking strategy alternates between these two methods based on a probability parameter $p_r$, with random masking applied $p_r$ of the time and interaction masking $(1 - p_r)$. To evaluate the effect of $p_r$, we test different values $0.7, 0.8, 0.9$ and find that $p_r = 0.8$ offers the best balance between interaction and reaction generation, as shown in Table 4. In the second stage, step unroll masking is applied which retains some of the predicted tokens from stage 1, remasks the remaining tokens and predicts them again. This is employed to incorporate the inference-time progressive refinement of tokens in the training process.

# E IMPLEMENTATION DETAILS

Our models are implemented using PyTorch, with details of the model architecture, training, and inference provided below. Key hyperparameters are summarized in the accompanying tables.

## E.1 MODEL ARCHITECTURE

The Motion VQ-VAE employs 2D convolutional residual blocks for both the encoder and decoder. The temporal downsampling factor is $n/N = 1/4$ for both datasets, while the spatial downsampling is dataset-specific: $j/J = 5/22$ for InterHuman and $j/J = 5/56$ for InterX. Strided convolutions are used for downsampling in the encoder, while the decoder restores dimensions via upsampling and convolutional layers. The latent representation in VQ-VAE has a dimension $d' = 512$, and the codebook size $|\mathcal{C}| = 1024$.

For the Inter-M transformer, we use $\mathbf{L} = 6$ transformer blocks, each with 6 attention heads. The transformer embedding dimension is $\tilde{d} = 384$.

## E.2 TRAINING DETAILS

The Motion VQ-VAE is trained for 50 epochs with a batch size of 512. The learning rate is initialized at 0.0002 and decays via a multistep learning rate schedule, reducing by a factor of 0.1 after 70% and 85% of the iterations. A linear warm-up is applied for the first quarter of the iterations. The

| Parameter | Value | Description |
|---|---|---|
| $d'$ | 512 | Latent space dimension of VQ-VAE |
| $|\mathcal{C}|$ | 1024 | Codebook size (number of entries) |
| $n/N$ | 1/4 | Temporal downsampling factor for both datasets |
| $j/J$ (InterHuman) | 5/22 | Spatial downsampling for InterHuman dataset |
| $j/J$ (InterX) | 5/56 | Spatial downsampling for InterX dataset |
| **L** | 6 | Number of transformer blocks |
| Attention heads | 6 | Number of attention heads per block |
| $\tilde{d}$ | 384 | Transformer embedding dimension |
| CLIP version | ViT-L/14@336px | Version of CLIP used for text in transformer |

Table 5: Motion VQ-VAE and Inter-M Transformer **Model Parameters**

commitment loss factor $\beta$ is 0.02, and the geometric losses for velocity, foot contact, and bone length are weighted differently across the datasets.

The Inter-M transformer is trained for 500 epochs with a batch size of 52, following a similar multi-step learning rate decay but with a decay factor of 1/3 after 50%, 70%, and 85% of the iterations. The condition drop probability is 0.1 to allow for flexibility in training with or without text conditioning.

| Parameter | Value | Description |
|---|---|---|
| VQ-VAE batch size | 512 | Number of samples per batch for VQ-VAE |
| Transformer batch size | 52 | Number of samples per batch for transformer |
| Initial learning rate | 0.0002 | Starting learning rate for both models |
| Learning rate decay | 0.1 / 1/3 | Decay factor for VQ-VAE / Transformer learning rate |
| $\beta$ | 0.02 | Commitment loss factor for VQ-VAE |
| $\lambda_{vel}, \lambda_{fc}, \lambda_{bl}$ (InterHuman) | 100, 500, 5 | Geometric loss weights for InterHuman |
| $\lambda_{vel}, \lambda_{fc}, \lambda_{bl}$ (InterX) | 100, 100, 5 | Geometric loss weights for InterX dataset |
| Condition drop prob. | 0.1 | Drop probability for text conditioning during transformer training |
| $p_r$ | 0.8 | Random Masking probability for stage 1 masking during training |

Table 6: **Training Hyperparameters** for the Motion VQ-VAE and Inter-M Transformer

### E.3  INFERENCE DETAILS

During inference, the number of iterations $I$ is set to 20 for interaction generation and 12 for reaction generation. A classifier-free guidance (CFG) scale of 2 is applied, and the temperature is set to 1 to balance diversity and coherence in the generated results.

| Parameter | Interaction Generation | Reaction Generation |
|---|---|---|
| Number of iterations | $I = 20$ | $I_{react} = 12$ |
| CFG scale | 2 | |
| Temperature | 1 | |

Table 7: **Inference Hyperparameters** for Interaction and Reaction Generation

## F  DIVERSITY DEMONSTRATION

Our quantitative comparison (section 4.1) shows that InterMask prioritizes adherence to text over extreme diversity, while still being able to generate different distinct interactions for the same text prompt. Here, in Figure 9, we show 2 visual examples to demonstrate this. In both cases, the model remains consistent in generating interactions described in the text prompt while exhibiting distinct features in different samples. For the *dancing* case, first sample shows waving hands in the beginning followed by a synchronized forward step, while individuals face the same direction throughout. The second sample shows individuals facing each other in the beginning, followed by waving hands and

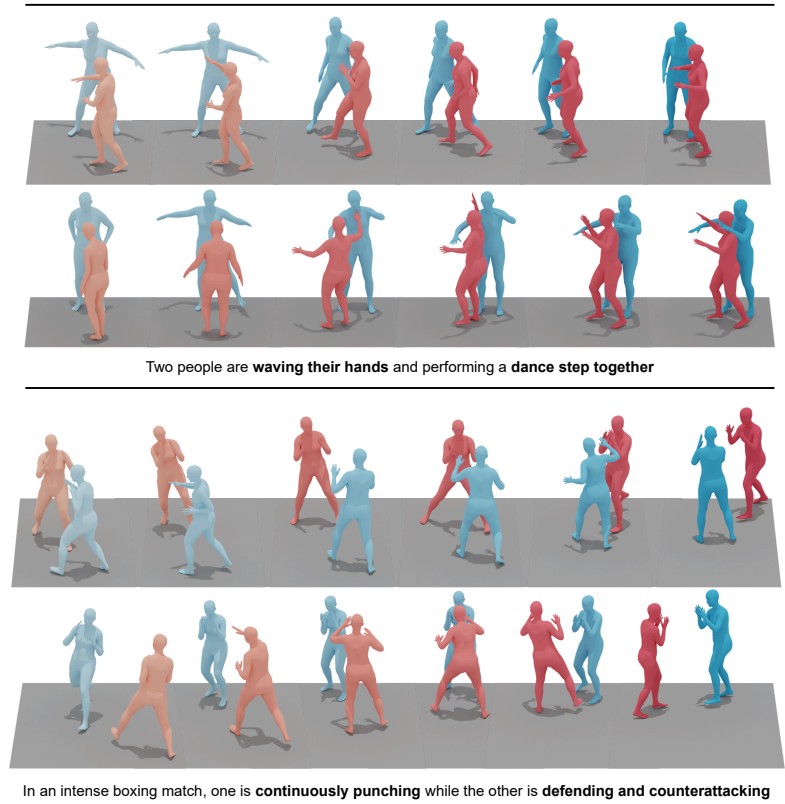

Figure 9: Diversity Demonstration of our method, where it generates two distinct interaction sequences for the same text prompt.

a synchronized spin. For the *boxing* case, first sample shows one individual punching three times and continuously moving forward, while the second sample shows them punching two times and retreating at the end.

# G   ALTERNATIVE MODELING

In our ablation study (section 4.2), we explore a one-at-a-time modeling framework for interaction generation, referred to as *Alternative Modeling*, which contrasts with the collaborative modeling framework of InterMask. While collaborative modeling predicts the tokens of both individuals simultaneously, alternative modeling follows a sequential process, generating tokens for one individual at a time, conditioned on the thus far predicted tokens of the other.

During training, as shown in Figure 10(a), we randomly mask both individuals' tokens $\{\tilde{t}_a, \tilde{t}_b\}$ and obtain their embeddings $\{\mathbf{e}_a, \mathbf{e}_b\}$ through the input process. Then, only the tokens of one individual are predicted ($\hat{t}_a$) by passing their embeddings through the transformer blocks, conditioned on the embeddings of the other individual using a cross-attention module. During inference (Figure 10(b)), both individuals' tokens are initially fully masked $\{t_a(0), t_b(0)\}$. In the first iteration, the tokens of one individual are predicted, and these are remasked based on their confidence scores to obtain $t_a(1)$. The second individual's tokens are then predicted in the next iteration, conditioned on the retained tokens from the first, to obtain $t_b(1)$. This alternation continues iteratively, progressively refining the tokens of both individuals. As shown in Table 3, the FID score for alternative modeling is 7.637 (compared to 5.154 for collaborative modeling), and the R-precision is 0.340 (compared to 0.449). These results indicate that while alternative modeling increases diversity, collaborative modeling produces high-quality and more realistic interactions, offering a better balance between diversity and interaction fidelity.

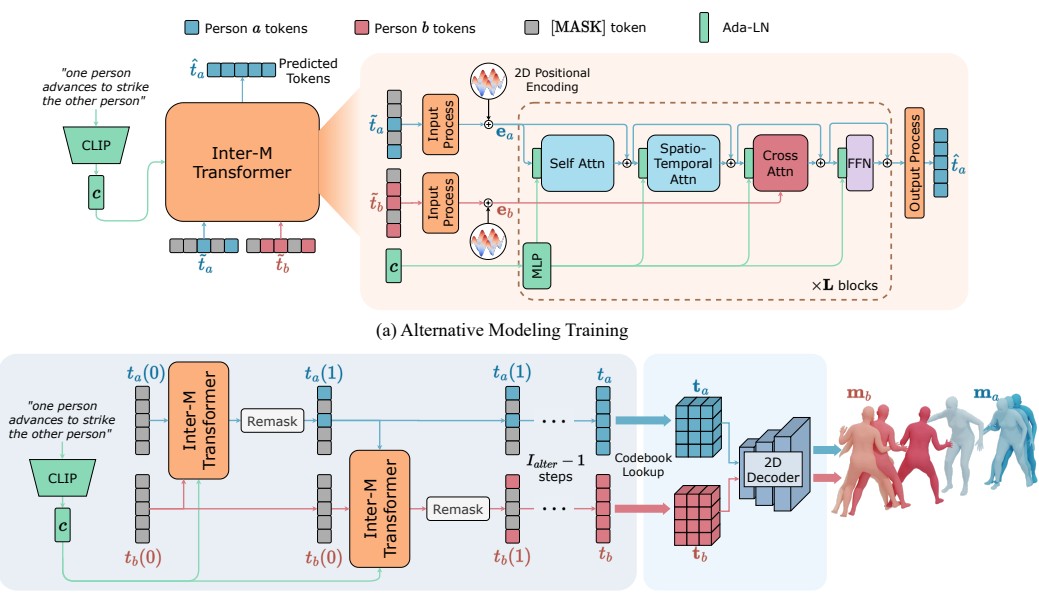

(a) Alternative Modeling Training

(b) Alternative Modeling Inference

Figure 10: Overview of the **Alternative Modeling** approach, where we predict the tokens of one person at a time. (a) During training, only the embeddings of one individual $\mathbf{e}_a$ are updated in the transformer blocks, conditioned on the other individual's embeddings $\mathbf{e}_b$. (b) During inference, the process alternates between predicting and remasking the tokens of each individual, starting with both fully masked $\{t_a(0), t_b(0)\}$. This process continues for $I_{alter}$ iterations for each individual.

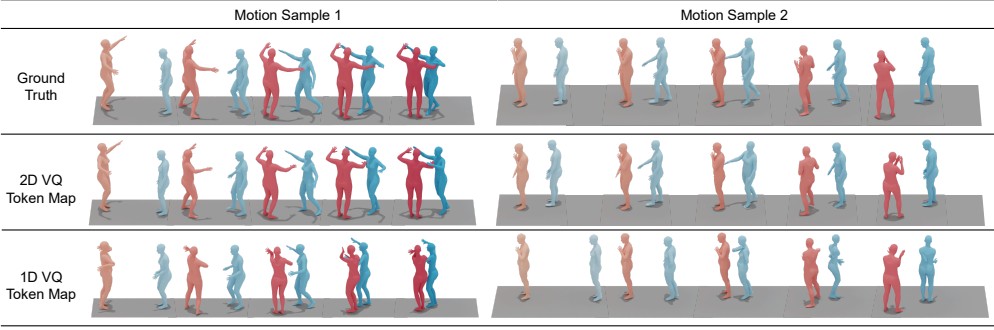

Figure 11: Qualitative results for the ablation study on Motion VQ-VAE to verify the proposed 2D token map.

## H   ABLATION STUDY QUALITATIVE RESULTS

In this section, we present qualitative results for our ablation studies to complement the quantitative findings discussed in section 4.2. Figure 11 shows two ground truth interaction sequences with their reconstructed samples from the 2D token map VQ-VAE and the baseline 1D token map VQ-VAE. As shown, the 1D VQ-VAE struggles to accurately reconstruct the spatial positions and orientations of the joints for both individuals, leading to incorrect positioning and orientation relative to each other. This results in not only unrealistic interactions but also bizarre individual poses. In contrast, the proposed 2D VQ-VAE provides highly accurate reconstructions at both the individual pose and the collective interaction level.

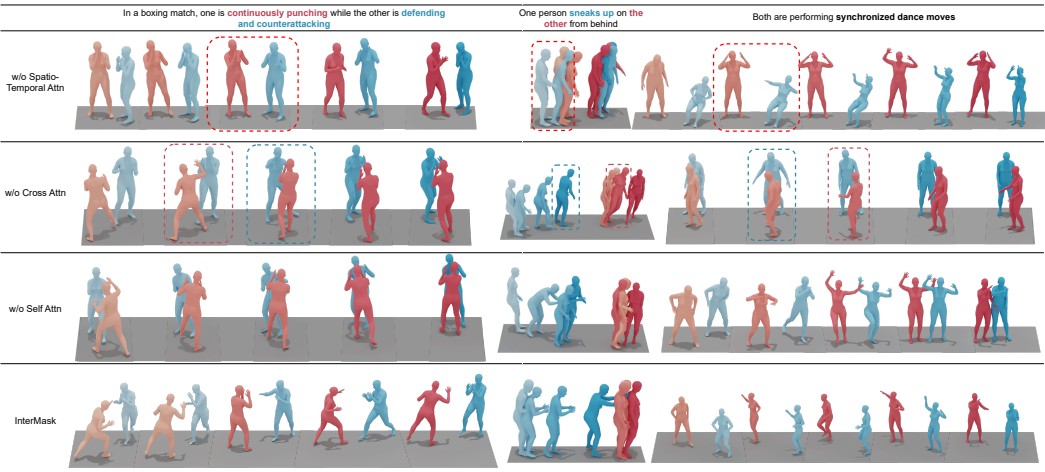

Figure 12: Qualitative results for the ablation study on Inter-M Transformer to verify contributions of the proposed Attention modules.

Figure 12 illustrates the impact of different attention modules in our Inter-M transformer by comparing outputs by removing the spatio-temporal attention, cross-attention, and self-attention modules independently. We provide results for three distinct interaction scenarios: *boxing*, *sneaking*, and *synchronized dancing*. The spatio-temporal attention module emerges as a critical component for generating complex poses and ensuring spatial awareness in interactions. Without this module, the *boxing* scenario exhibits overly simplistic poses, such as timid *punching* and *blocking*, alongwith the individuals failing to face each other properly. In the *sneaking* scenario, the absence of spatio-temporal attention eliminates the essential spatial progression, as the sneaking individual does not gradually approach the other. Similarly, in the *dancing* scenario, the generated motions are reduced to basic hand raises, and one individual adopts an unnatural crouching pose. By contrast, the inclusion of spatio-temporal attention enables accurate spatial positioning and expressive, synchronized interactions. The cross-attention module seems vital for modeling inter-person dependencies, particularly in achieving accurate reaction timing. Without it, the response motions of the interacting individual appear either delayed or prematurely executed across all examples. For instance, in the *boxing* scenario, the reactive movements fail to synchronize with the initiating individual's punches. In the *dancing* scenario, the lack of cross-attention results in poor synchronization, disrupting the fluidity of the interaction. Lastly, the self-attention module serves as a refinement mechanism, enhancing the overall quality and coherence of individual motions. Its removal introduces subtle inconsistencies, such as jerky transitions or less fluid movements, which slightly degrade the interaction's realism.These observations collectively underscore the importance of each attention module in generating realistic, contextually accurate, and expressive interactions.

## I    REACTION GENERATION INFERENCE

InterMask does not require task-specific fine-tuning or architectural re-design for reaction generation, needing only minor adjustments to the inference process, as illustrated in Figure 13. The process begins by encoding the reference individual's motion $\mathbf{m}_b$ into tokens $t_b$ using the VQ-VAE encoder. For the other individual, whose reaction is to be generated, we initialize with a fully masked token sequence, $t_a(0)$. Over the course of $I_{\text{react}}$ iterations, the transformer progressively predicts and fills in the masked tokens, while the reference tokens remain unmasked throughout. At each iteration $i_{\text{react}}$, the least confident $\gamma \left( \frac{i_{\text{react}}}{I_{\text{react}}} \right) \cdot nj$ tokens are remasked and predicted again, following a cosine scheduling function $\gamma(\cdot)$. Once all tokens are generated, the final token sequence $t_a$ is decoded back into motion $\mathbf{m}_a$ using the VQ-VAE decoder. Since we drop the conditioning signal during some training passes, reaction generation functions effectively both with and without a text description, enabling the model to generate motions based solely on the reference motion or guided by additional textual instructions.

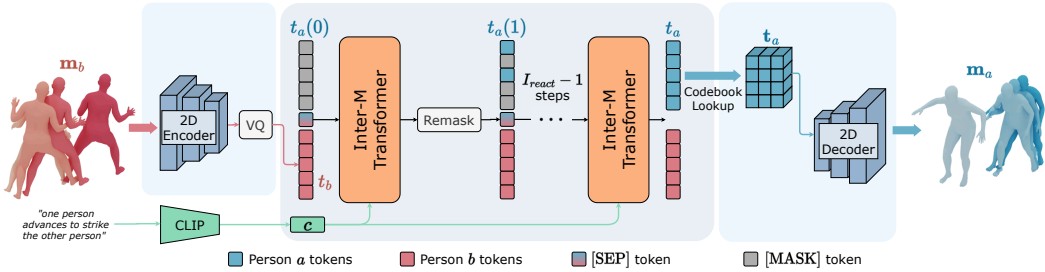

Figure 13: **Inference** process for the **Reaction Generation** task. The motion tokens of the reference individual $t_b$ are obtained from the encoder and kept unmasked throughout. The second individual's tokens are initially fully masked $t_a(0)$, and are predicted progressively over $I_{react}$ iterations to obtain $t_a$, which is then decoded using the decoder.

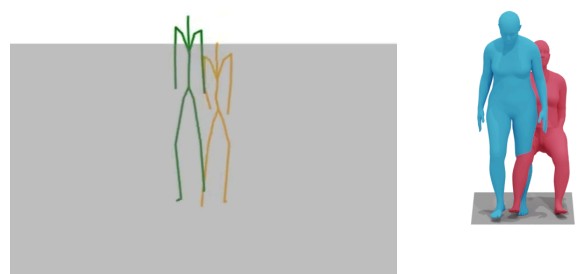

First person is **sitting in a chair**, the second **takes a step forward with their right foot**

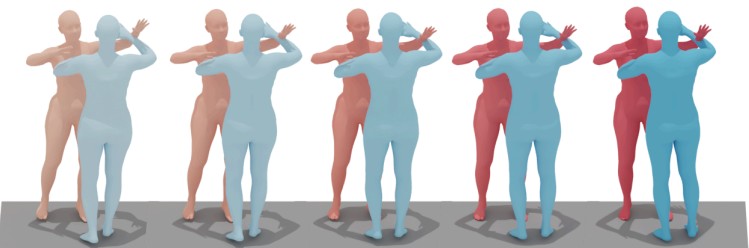

The first takes a **step with their left foot**

Figure 14: Examples of Limitations of our method. The first row shows body penetration when converted from output skeleton to SMPL mesh. The second row shows implicit bias towards dancing.

## J   FAILURE CASES

In Figure 14, we show visual examples of failure cases emerging from the limitations of our method, as decribed in section 5. In the first row, we show that when converting our output skeletons to SMPL (Loper et al., 2015) meshes for visualizations, the results can exhibit body penetration among the interacting individuals. One possible future solution to this problem is to include the mesh conversion in the training process and incorporate anti-penetration in the training loss. In the second row, we show that the model suffers from some implicit biases present in the dataset, where it assumes that the individuals are dancing without explicit mention in the text prompt.

# K    USER STUDY

We conducted the user study on the Amazon Mechanical Turk platform, where the interface presented to the users is shown in Figure 15. For each sample, users were provided with clear instructions to rate two animations—one generated by InterMask and the other by InterGen (Liang et al., 2024)—from the same text description, with the order of the animations randomized for each sample to avoid bias. Participants were asked to rate both animations on a scale of 1 to 5 for interaction quality and text adherence. Following the individual ratings, they were asked to select the better animation based on their overall impression. A total of 16 users evaluated a total 30 samples, with each sample being rated by three users. To ensure high-quality feedback, we filtered users to include only those with Amazon Mechanical Turk *master* status, with a task approval rating of over 97% and more than 1000 previously approved tasks.

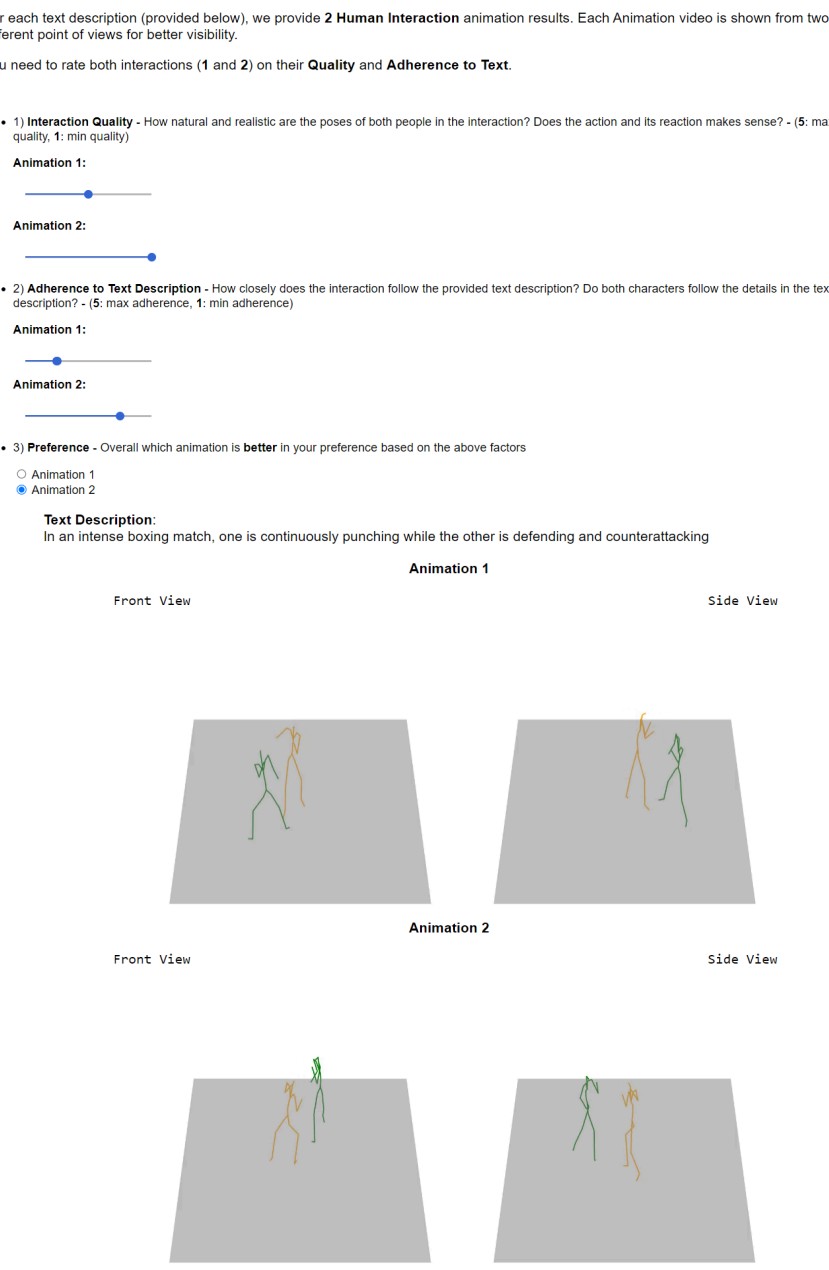

Figure 15: Interface of the **User Study** on Amazon Mechanical Turk.

