# OpenReview forum: "InterMask: 3D Human Interaction Generation via Collaborative Masked Modeling"
_ICLR.cc/2025/Conference — ICLR 2025 Poster_

### Official Review · Reviewer_YC4H · 2024-11-02

**Soundness:** 3
**Presentation:** 3
**Contribution:** 2
**Rating:** 6
**Confidence:** 5

**Summary:**

This paper addresses the problem of generating human-human interaction from textual descriptions. The authors introduce InterMask, a novel framework that employs 2D VQ-VAE to encode interactions into a discrete space and generative masked modeling to capture complex spatio-temporal interdependencies.

**Strengths:**

1. This paper introduces a novel 2D VQ-VAE that quantizes motion into a 2D discrete map, which can more effectively preserve fine-grained spatio-temporal details than the 1D version.

2. The proposed method achieves SOTA performance on the InterHuman and InterX benchmarks.

**Weaknesses:**

1. The method does not appear substantially different from prior work. Overall, the paper feels somewhat underwhelming, lacking distinctive insights, though there are no major issues with the experiments or writing. I would rate it as borderline.

2. The proposed approach faces challenges in generating human reactions in an online manner (although this isn’t the primary focus of the current setting). This capability would be more practical for some virtual reality applications.

**Questions:**

I have no other questions.

---

> ### Author Response · Authors · 2024-11-21
> **Reply to Reviewer YC4H**
>
> **W1**: *The method does not appear substantially different from prior work. Overall, the paper feels somewhat underwhelming, lacking distinctive insights, though there are no major issues with the experiments or writing. I would rate it as borderline.*
>
> **Ans**: We thank the reviewer for their feedback and appreciate the opportunity to clarify the key contributions of our work. Our method introduces several novel insights and improvements that set it apart from prior work:
>
> - **Masked Modeling for Interaction Generation**: Unlike previous approaches, we are the first to model interactions in the discrete space leveraging a generative masked modeling framework specifically designed for two-person interaction generation.
> - **Novel 2D Motion Token Map**: We propose a 2D token map representation in the VQ-VAE, which preserves both temporal and spatial dimensions. This is a significant advancement over prior methods that relied on 1D VQ-VAEs, as it captures finer-grained spatio-temporal details crucial for modeling interactions.
> - **Inter-M Transformer**: Our architecture features a dedicated Inter-M Transformer designed to collaboratively model the 2D tokens of two individuals. This includes custom attention modules and complementary masking strategies that enhance learning of both intra- and inter-person dependencies. Additional visual ablation studies for the attention modules are added in the Appendix F of the paper and Section C of the supplementary video webpage.
> - **Flexibility for Reaction Generation and Beyond**: Our framework is the first that supports reaction generation without requiring any task-specific fine-tuning, showcasing its flexibility. Furthermore, it has the potential to be adapted for multi-person interaction scenarios with minimal architectural modification, only to the cross attention module. This module would need to condition each individual’s tokens on a joint representation of all other participants’ tokens, rather than just one partner.
>
> We believe these contributions collectively represent a meaningful advancement in the field of interaction generation and hope this clarification addresses the reviewer’s concern.
>
> ---
>
> **W2**: *The proposed approach faces challenges in generating human reactions in an online manner (although this isn’t the primary focus of the current setting). This capability would be more practical for some virtual reality applications.*
>
> **Ans**: We thank the reviewer for highlighting the challenge of generating human reactions in an online manner. This capability is indeed more relevant for the reaction generation task, which is not the primary focus of our work. Our main focus lies in generating human interactions simultaneously, a setting applicable to two interacting digital humans in scenarios such as animation for films, games, and virtual reality, as well as robotics applications requiring pre-designed motion plans for multi-agent interactions.
>
> That said, we do demonstrate reaction generation in an offline setting as an additional application of our method. Our approach can seamlessly handle this task without requiring task-specific fine-tuning or modifications, showcasing the flexibility of our framework. While online reaction generation is a promising direction, it lies outside the scope of this work, and we look forward to exploring it in future research.
>
> ---
>
> *Thank you so much for reviewing our work. We sincerely hope that we have properly addressed your concerns. If not, we are happy to open further discussions.*

---

> ### Author Response · Authors · 2024-12-02
> **Additional Response to Reviewer YC4H**
>
> Dear Reviewer YC4H,
>
> We would like to express our sincere thanks again for your thoughtful feedback and for highlighting the strengths of our work, including the novel 2D discrete motion token map, effective preservation of fine-grained spatio-temporal details, and our state-of-the-art performance on the InterHuman and InterX benchmarks.
>
> We understand your concern about not being substantially different from prior work. We would like to take this opportunity to reiterate the key distinctions and insights of our work, and kindly request if you would consider re-evaluating our work. First, we believe that our introduction of a 2D discrete motion token map marks a significant contribution, as this representation has not previously been explored in the domain of human motion modeling. While it is particularly impactful for interaction generation, where heightened spatial awareness is crucial, its potential applications extend to a wide range of motion modeling directions, including single-person motion generation, one-to-many motion generation, and human-scene interactions. Second, we propose the use of generative masked modeling in place of previous diffusion models for interaction generation. This novel paradigm, combined with appropriate tokenization and attention mechanisms, significantly improves performance while maintaining lower inference time and offering greater flexibility in handling complimentary tasks such as reaction generation.
>
> We sincerely hope our responses and the recent additions made to the paper and supplementary materials adequately address your concerns, and we would really appreciate it if you would consider re-evaluating our work. As a gentle reminder, *the discussion period will end in about two days, with the reviewer response period concluding in approximately 30 hours*. We would love to hear back from you and are happy to discuss any further questions or concerns you might have. Thank you again for your time and thoughtful engagement with our work.

---

> ### Comment · Reviewer_YC4H · 2024-12-02
>
> Thank you for your response. I have no further concerns. While the paper is suitable as a poster, it fails to convey exciting insights, so I will maintain my original rating.

---

### Official Review · Reviewer_LgJ2 · 2024-11-03

**Soundness:** 3
**Presentation:** 2
**Contribution:** 3
**Rating:** 6
**Confidence:** 4

**Summary:**

The paper presents InterMask, a framework for generating 3D human interactions using masked modeling in discrete space. By encoding motion as 2D token maps and modeling intra- and inter-person dependencies with a custom transformer, InterMask achieves impressive results on the InterHuman and InterX datasets. The model shows strong performance in terms of realism and alignment with text prompts, outperforming existing methods in controlled settings.

**Strengths:**

1. The use of 2D token maps and spatial-temporal attention layers in the Inter-M transformer enables more nuanced modeling of human interactions, demonstrating improved fidelity in motion generation.

2. InterMask achieves state-of-the-art results on key metrics in standard datasets, indicating its effectiveness in generating realistic, text-aligned interactions.

3. The model is relatively lightweight and demonstrates reduced inference time.

**Weaknesses:**

1. The paper’s innovative contribution is not well-defined. Although the authors introduce a transformer module to model intra- and inter-person spatial-temporal dependencies, this approach is widely used in various generative tasks. The ablation studies, which compare only with/without this module, do not sufficiently demonstrate the model’s unique contribution. If this module is intended as a core innovation, the authors should focus on explaining why its internal structure, arrangement, or sequence specifically suits the 3D interaction modeling problem. Merely showing the presence or absence of the module does not justify its relevance, as it is a fairly standard approach.

2. The process of converting each motion sequence into a 2D discrete motion token map lacks detailed description. The main innovation of this paper appears to lie in the combination of the 2D discrete motion token map with MAE; however, there is insufficient explanation of how the 2D token map is constructed. Providing a more thorough description, ideally with a detailed diagram, would enhance clarity and better communicate this aspect of the method.

3. User Study. The user study includes only three participants, which is insufficient to draw robust conclusions on user preference. A larger sample size and detailed descriptions of the study’s methodology would add reliability and provide stronger validation for claims of user preference for InterMask-generated interactions.

4. Upon reviewing the supplementary videos, the generated motions appear to lack the fluidity seen in InterGen’s outputs. A discussion on this aspect would be valuable, particularly to understand any limitations in InterMask’s approach to generating smooth, natural motion.

**Questions:**

1. Could the authors comment on the suitability of the current framework for multi-person interactions? Given the focus on two-person setups, it would be useful to understand any potential modifications or challenges in adapting this approach for more complex, multi-person interaction scenarios.

2. How does the model handle longer, more complex instructions or instructions that are less structured (in-the-wild)?

3. Do the authors see potential for combining InterMask with language models to improve interpretability or to handle a wider range of textual prompts?

---

> ### Author Response · Authors · 2024-11-22
> **Reply to Reviewer LgJ2 (1/3)**
>
> **We thank the reviewer for their detailed review and thought-provoking discussion, and hope the following response addresses your concerns adequately.**
>
> **W1**: *The paper’s innovative contribution is not well-defined. Although the authors introduce a transformer module to model intra- and inter-person spatial-temporal dependencies, this approach is widely used in various generative tasks. The ablation studies, which compare only with/without this module, do not sufficiently demonstrate the model’s unique contribution. If this module is intended as a core innovation, the authors should focus on explaining why its internal structure, arrangement, or sequence specifically suits the 3D interaction modeling problem. Merely showing the presence or absence of the module does not justify its relevance, as it is a fairly standard approach.*
>
> **Ans**: We thank the reviewer for their comments and appreciate the opportunity to clarify the innovative contributions of our work. Our main contribution lies in being the first to model human interactions in the discrete space using generative masked modeling. The core novelty of our approach is the discrete modeling of human interactions and the introduction of the novel 2D motion token map, which captures both spatial and temporal dimensions. The effectiveness of these contributions is evidenced by the VQ-VAE ablation results, where the 2D token map outperforms its 1D counterparts, and the overall performance of the framework, outperforming previous SOTA methods.
>
> While the design of the Inter-M Transformer is not a novel contribution, it plays an important role in enabling the collaborative modeling of the 2D tokens for two interacting individuals. Its internal structure combines self-attention, spatio-temporal attention, and cross-attention mechanisms to address the complexities of interaction generation. This design ensures the modeling of intra- and inter-person dependencies in a manner well-suited for 3D interaction scenarios, setting it apart from generic transformer architectures. To further justify the relevance and effectiveness of this design, we conducted additional visual ablation experiments, complementing the quantitative ablations in Section 4.2. As detailed in Appendix F, these visual results demonstrate the specific contributions of each attention mechanism in different interaction scenarios, such as boxing, synchronized dancing, and sneaking up. The spatio-temporal attention module is crucial for handling complex poses and spatial awareness, the cross-attention mechanism ensures accurate and temporally synchronized reactions, and the self-attention module refines the overall quality. These findings highlight how the Inter-M Transformer’s structure is specifically tailored to address the challenges of 3D interaction modeling. We have also added the video ablation results to Section C of the supplementary video webpage.
>
> ---
>
> **W2**: *The process of converting each motion sequence into a 2D discrete motion token map lacks detailed description. The main innovation of this paper appears to lie in the combination of the 2D discrete motion token map with MAE; however, there is insufficient explanation of how the 2D token map is constructed. Providing a more thorough description, ideally with a detailed diagram, would enhance clarity and better communicate this aspect of the method.*
>
> **Ans**: We thank the reviewer for their feedback and for pointing out the need for a more detailed explanation of the 2D discrete motion token map construction. In our VQ-VAE, the 2D encoder processes individual motion sequences represented as $N\times J\times d$, where $N$ is the number of poses, $J$ is the number of joints, and $d$ is the joint feature dimension. The encoder uses 2D convolutional layers to down-sample the motion along temporal and spatial dimensions, producing a latent representation with reduced dimensions, $n\times j\times d’$. This representation is then quantized using a learned codebook, where each feature vector is replaced with the index of its nearest codebook entry. The result is a 2D motion token map, capturing both spatial and temporal contexts for each token.
>
> To address the reviewer's concern, we have added a detailed explanation and an accompanying diagram in Appendix J of the paper. This section includes the internal structure of the 2D encoder, the quantization process, and the overall workflow for generating the token map. We hope this addition clarifies the construction of the 2D token map and enhances the paper's accessibility.
>
> ---

---

> ### Author Response · Authors · 2024-11-22
> **Reply to Reviewer LgJ2 (2/3)**
>
> **W3**: *User Study. The user study includes only three participants, which is insufficient to draw robust conclusions on user preference. A larger sample size and detailed descriptions of the study’s methodology would add reliability and provide stronger validation for claims of user preference for InterMask-generated interactions.*
>
> **Ans**: We thank the reviewer for their comment and apologize for the confusion regarding the user study methodology. To clarify, the user study consisted of a total of 16 participants, where each sample was independently evaluated by three different participants to ensure diverse feedback. Hence each participant, on average, reviewed two samples out of the total 30 interaction samples. This setup resulted in a total of 90 evaluation data points (30 samples × 3 evaluations per sample), providing a robust foundation for assessing user preferences. We believe this approach, with multiple participants evaluating each sample and a sufficient number of total evaluations, adds reliability to our findings and supports the conclusion that InterMask-generated interactions are preferred over baseline methods. We have rectified this confusion in the paper script as well.
>
> ---
>
> **W4**: *Upon reviewing the supplementary videos, the generated motions appear to lack the fluidity seen in InterGen’s outputs. A discussion on this aspect would be valuable, particularly to understand any limitations in InterMask’s approach to generating smooth, natural motion.*
>
> **Ans**: We thank the reviewer for their observation regarding the fluidity of generated motions. The lack of fluidity in some outputs arises from the SMPL conversion process, where we used the MotionGPT [1] code to convert one frame at a time. This frame-by-frame approach can introduce temporal discontinuities, making the motions appear less smooth. However, our model’s outputs at the joint level are temporally consistent and smooth. To demonstrate this, we have included supplementary videos in the Section A of the webpage comparing the smooth joint-level outputs of our model with the SMPL-converted animations side by side. The sudden or disconnected movements observed are artifacts of the conversion process, not of our underlying model. Future work will explore improved conversion algorithms that account for temporal consistency to generate smoother outputs.
>
> ---
>
> **Q1**: *Could the authors comment on the suitability of the current framework for multi-person interactions? Given the focus on two-person setups, it would be useful to understand any potential modifications or challenges in adapting this approach for more complex, multi-person interaction scenarios.*
>
> **Ans**: We thank the reviewer for raising this important question regarding the suitability of our framework for multi-person interactions. Our framework can be extended to handle multi-person scenarios with minimal modifications due to its modular design. Most components, including the VQ-VAE, self-attention, and spatio-temporal attention modules, can be used without changes. The VQ-VAE can encode the motion of each individual into tokens, which can then be concatenated with $[SEP]$ tokens to separate individuals before being processed by the self-attention module. The spatio-temporal attention module can still model intra-person dependencies independently for each individual.
>
> The primary modification required lies in the cross-attention mechanism. For multi-person interactions, this mechanism would need to condition each individual’s tokens on a joint representation of all other participants’ tokens, rather than just one partner. With this adjustment, our framework can effectively scale to model more complex multi-person interaction scenarios while retaining its existing strengths.
>
> ---
>
> [1] MotionGPT: Human Motion as a Foreign Language, NeurIPS 2023

---

> ### Author Response · Authors · 2024-11-22
> **Reply to Reviewer LgJ2 (3/3)**
>
> **Q2**: *How does the model handle longer, more complex instructions or instructions that are less structured (in-the-wild)?*
>
> **Ans**: We thank the reviewer for raising this important question. We conducted additional experiments with longer, more complex, and unstructured textual instructions, and have added them to Section D of the supplementary video webpage.
>
> For complex instructions involving multiple steps in progression and alternating actions between two individuals, our model performs well, as demonstrated in the first two examples. However, for more out-of-distribution texts that the model did not encounter during training, it interprets cues as best as possible to generate plausible interactions. For instance, while the model does not fully understand "pointing a gun," it generates a sample where one person points at the other, who raises their hands. Similarly, in the "Goku vs. Vegeta" scenario, the model understands the context of a fight, producing karate-like poses but not specific moves like "kamehameha." For prompts like the "Fortnite Orange Justice dance," it generates a celebratory dance with two winners but does not replicate the specific moves.
>
> These limitations highlight the need for future work, which could incorporate foundational models of language, motion, or multimodal representations, utilize additional single-person motion data, or expand interaction datasets, potentially sourced from internet videos. As described earlier, incorporating pre-trained language models (LLMs) could enhance the framework’s ability to process and interpret complex and in-the-wild instructions as well.
>
> ---
>
> **Q3**: *Do the authors see potential for combining InterMask with language models to improve interpretability or to handle a wider range of textual prompts?*
>
> **Ans**: We thank the reviewer for this insightful question. Incorporating language models (LLMs) into InterMask offers promising opportunities to enhance interpretability and handle a broader range of semantic concepts. One potential approach is to use a pre-trained LLM, to refine and expand the textual prompts. The LLM could generate more structured, detailed or enriched descriptions of interactions, which would then serve as more informative conditioning inputs for the Inter-M Transformer. This could improve the alignment between textual prompts and generated interactions, especially for ambiguous (in-the-wild) or complex prompts.
>
> Another avenue is to incorporate motion tokens into the “language” of a pre-trained LLM. This could be achieved through fine-tuning or adaptation techniques like LoRA [2], enabling the LLM to process motion tokens alongside textual inputs. By unifying the representation of text and motion, this integration could improve the framework’s ability to handle more complex prompts and generate interactions with enhanced coherence and diversity. Another potential approach is to replace the current CLIP-based text representation with embeddings obtained from a pre-trained LLM. These embeddings could provide richer semantic representations, enabling the model to better capture nuances in textual prompts and align them with the generated interactions.
>
> We believe these directions have significant potential and look forward to seeing their implications in future work.
>
> ---
> [2] LoRA: Low-Rank Adaptation of Large Language Models, ICLR 2022
>
> *We sincerely thank you for taking the time to review our work. We have made our best effort to address your concerns, and hope we have done so adequately. Please feel free to let us know if you have further questions.*

---

> ### Comment · Reviewer_LgJ2 · 2024-11-26
> **Official Comment by Reviewer LgJ2**
>
> I appreciate the author's response and the extensive additional demonstrations, which have addressed most of my concerns. I remain positive about the paper, leaning towards acceptance.

---

> > ### Author Response · Authors · 2024-12-02
> > **Additional Response to reviewer LgJ2**
> >
> > Dear Reviewer LgJ2,
> >
> > We would like to thank you again for your thoughtful feedback and for highlighting the strengths of our work, including the novel 2D discrete motion token map and the use of spatio-temporal attention for more nuanced modeling of human interactions, our state-of-the-art results on standard datasets, and the significantly reduced inference time. We are also grateful for the areas of improvement you pointed out, which have helped us enhance the soundness, contribution, and presentation of our paper. We deeply appreciate your acknowledgment that our efforts have addressed your concerns and kindly ask if you would consider re-evaluating the score, if possible.
> >
> > As a gentle reminder,  *the discussion period will end in about two days, with the reviewer response period concluding in approximately 32 hours*. If you have any additional concerns or thoughts, we would love to hear from you and discuss further. Thank you again for your time and constructive engagement with our work.

---

### Official Review · Reviewer_jS9r · 2024-11-03

**Soundness:** 2
**Presentation:** 2
**Contribution:** 2
**Rating:** 6
**Confidence:** 3

**Summary:**

This article introduces a new framework called InterMask for generating 3D human interaction actions from text descriptions. The proposed framework appears to be based on the T2M-GPT framework, modeling in discrete space using generative masks, while also incorporating the concept of interhand to collaborate within the mask modeling framework by utilizing tokens from two interacting individuals. During the inference process, it starts with a fully masked sequence and gradually fills in the tokens for both individuals.

**Strengths:**

1. A Transformer structure combined with VQ-VAE has been proposed for generating 3D human interaction actions.
2. It utilizes a 2D discrete motion token map to represent motion, distinguishing itself from traditional 1D representation methods. This representation better preserves spatiotemporal details and spatial awareness, which is crucial for accurately capturing complex motion relationships in human interactions.
3. The framework has achieved state-of-the-art results on the InterHuman and InterX datasets, performing exceptionally well on various metrics.
4. The average inference time is only 0.77 seconds, significantly lower than InterGen's 1.63 seconds, providing a considerable advantage in practical applications by enabling faster result generation.

**Weaknesses:**

1. In the visualization results, issues such as body penetration between interacting individuals and unnatural sliding movements still occur.
2. The article also mentions the masking strategy, but I believe its effectiveness may be influenced by various factors, such as the setting of the mask ratio and the dynamic adjustment of masks at different stages. The article does not delve deeply into how these factors affect model performance or how to optimally set these parameters.

**Questions:**

1. I noticed that in the second video of the visualization "Everyday Actions," the character on the right makes a sudden arm-raising motion, which seems somewhat disconnected. Additionally, is there a 10-second video available for reference? I saw that the authors mentioned the ability to generate interactive results up to 10 seconds long.

2. In terms of quantitative experiments, it seems there is no comparison with PriorMDM, which also produces results for two-person interactions. Similarly, there doesn't appear to be a comparison with other VQ-VAE methods like T2M-GPT. I'm curious about their results, especially since they perform very well on single-person datasets. If T2M-GPT were given a reference action sequence for one individual as an additional condition, how would it perform in generating interactive individuals, particularly on interactive datasets?

3. Regarding the Inter-M Transformer structure, although there is some evidence in the ablation studies concerning the relative importance of different attention mechanisms, it may not fully clarify their specific roles and trade-offs in different interaction scenarios and data distributions.

---

> ### Author Response · Authors · 2024-11-21
> **Reply to Reviewer jS9r (1/3)**
>
> **We thank the reviewer for their valuable and constructive feedback, and hope the following response addresses your concerns adequately.**
>
> **W1**: *In the visualization results, issues such as body penetration between interacting individuals and unnatural sliding movements still occur.*
>
> **Ans**: We thank the reviewer for their valuable feedback and for highlighting the issues of body penetration and unnatural sliding movements. These limitations are acknowledged in Section 5 (Limitations and Future Work) of our paper, and we recognize their impact on the overall quality of the results. Below, we elaborate on their causes and how they can be addressed in future work.
>
> **Body Penetration**: Body penetration occurs due to the SMPL mesh conversion in post-processing, where the expanded body shapes of both individuals results in overlaps. This is a known challenge when using SMPL meshes for visualization. To mitigate this, SMPL conversion and body penetration losses can be incorporated directly into the training process of the Inter-M Transformer. By making dequantization differentiable, as done in TokenHMR [1], we can perform back-propagation from the penetration loss on SMPL meshes to the parameters on the Inter-M Transformer.
>
> **Sliding Movements**: We believe that the issue of sliding foot movements is primarily due to limited data availability for two-person interactions compared to single-person motion, resulting in weaker priors for realistic individual movements. In future work, this can be alleviated by augmenting training data with single-person motion datasets or by integrating physics-based post-processing to ensure realistic ground contact.
>
> Despite these limitations, our work introduces key innovations, such as enhanced spatio-temporal fidelity and adherence to text, which establish a solid framework for addressing these challenges in the future.
>
> ---
>
> **W2**: *The article also mentions the masking strategy, but I believe its effectiveness may be influenced by various factors, such as the setting of the mask ratio and the dynamic adjustment of masks at different stages. The article does not delve deeply into how these factors affect model performance or how to optimally set these parameters.*
>
> **Ans**: We thank the reviewer for their thoughtful feedback on the masking strategy. As described in Section 3.2.1 of our paper, our two-stage masking strategy combines random masking, interaction masking and step unroll masking. In the first stage, a random proportion of tokens from both or one of the individuals is masked, constituting random masking and interaction masking respectively. Random masking is chosen with a probability of $p_r$ (value can be found in Table 5, Appendix C.2) and interaction masking with $1 - p_r$.  Interaction masking is specifically designed to ensure that the model learns inter-person dependencies and to improve performance in the reaction generation task. In the second stage, step unroll masking is employed which retains some of the predicted tokens from stage 1, remasks the remaining tokens and predicts them again. This is employed to incorporate the inference-time progressive refinement of tokens in the training process.
>
> Regarding the factors influencing their effectiveness, we appreciate the reviewer highlighting this aspect. First, the stage 1 random masking ratio $p_r​$ was set higher than $0.5$, to ensure it was used more than interaction masking. This choice was motivated by the intuition that random masking promotes more holistic learning and ensures in expectation to cover every masking scenario, while interaction masking is an addition to promote the learning of inter-person dependencies, benefiting the reaction generation task. We experimented with $p_r = \{ 0.7, 0.8,0.9\}$ and found that $0.8$ offered the best trade-off between performance on interaction generation and reaction generation. The results of these experiments are shown in the Table below (*Italics* indicate best result, **Bold** indicates second). Additionally, the number of tokens remasked during step unroll masking was determined using the cosine schedule based on the optimal number of inference steps (20).
>
> We hope this explanation addresses the reviewer’s concern and provides clarity on our masking technique and its parameter choices. We have also added a section in the appendix of the paper (Appendix K) with details on the two stage masking technique, along with an illustration of the process, and the results table presented in this comment.
>
> | $p_r$ | Interaction Generation |  | Reaction Generation |  |
> |-----|----------------------|--|-------------------|--|
> |     | FID $\downarrow$ | R-Prec Top1 $\uparrow$  | FID $\downarrow$  | R-Prec Top1 $\uparrow$  |
> | 0.7 | 5.214 | 0.447 | *2.85* | *0.476* |
> | 0.8 | **5.154** | **0.449** | **2.99** | **0.462** |
> | 0.9 | *5.152* | *0.450* | 3.37 | 0.416 |
>
> ---
>
> [1] TokenHMR: Advancing Human Mesh Recovery with a Tokenized Pose Representation, CVPR 2024

---

> ### Author Response · Authors · 2024-11-21
> **Reply to Reviewer jS9r (2/3)**
>
> **Q1**: *I noticed that in the second video of the visualization "Everyday Actions," the character on the right makes a sudden arm-raising motion, which seems somewhat disconnected. Additionally, is there a 10-second video available for reference? I saw that the authors mentioned the ability to generate interactive results up to 10 seconds long.*
>
> **Ans**: We thank the reviewer for bringing up the sudden movements observed in some visualizations. This issue arises from the SMPL conversion process, where we utilized the MotionGPT [2] code, which processes one frame at a time during conversion. This frame-by-frame approach can lead to disconnected or abrupt movements in the SMPL animations. A better conversion algorithm that incorporates temporal consistency in the motion would address this issue and produce smoother videos. To clarify, we have added supplementary videos in Section A of the webpage, showcasing this phenomenon. These videos compare the smooth joint-level output of our model with the SMPL-converted animations side by side. It can be observed that the sudden movements appear only after the SMPL conversion. Additionally, as the reviewer inquired about longer examples, we have included 10-second video samples in the Section B of the supplementary video webpage to demonstrate our model's ability to generate longer interaction sequences. We hope these additions provide further clarity and address the reviewer’s concerns.
>
> ---
>
> **Q2**: *In terms of quantitative experiments, it seems there is no comparison with PriorMDM, which also produces results for two-person interactions. Similarly, there doesn't appear to be a comparison with other VQ-VAE methods like T2M-GPT. I'm curious about their results, especially since they perform very well on single-person datasets. If T2M-GPT were given a reference action sequence for one individual as an additional condition, how would it perform in generating interactive individuals, particularly on interactive datasets?*
>
> **Ans**: We thank the reviewer for their observations. Regarding PriorMDM, please note that it is referred to as ComMDM [3] in our paper, following the naming convention used in related works such as InterGen [4] and InterX [5] to ensure consistency in the literature. Comparisons with ComMDM are included in Table 1. For T2M-GPT [6], we conducted experiments to adapt it to the interaction generation setting by training its VQ-VAE on individual motions and modifying its transformer to generate two-person interactive motions simultaneously. The results, reported in the tables below, demonstrate its performance relative to our proposed method. The experiment described in the review corresponds to the reaction generation setting, where one individual’s motion is given as input, and the other is generated in response. Since our primary focus is on interaction generation rather than reaction generation, we did not explore this specific experiment with T2M-GPT.
>
> We first compare the VQ-VAE proposed in T2M-GPT with our 2D VQ-VAE in terms of the Reconstruction FID and MPJPE.
> | Method | Reconstruction FID $\downarrow$ | MPJPE $\downarrow$ |
> |--------|-------------------|--------|
> | T2M-GPT | 7.323 | 0.891 |
> | InterMask | 0.970 | 0.129 |
>
> Then we compare the generation performance of the autoregressive transformer proposed in T2M-GPT with our Inter-M Transformer.
> | Method | R-Prec Top1 $\uparrow$| R-Prec Top2 $\uparrow$ | R-Prec Top 3 $\uparrow$ | FID $\downarrow$ | MMDist $\downarrow$ | Div $\rightarrow$ | MModality $\uparrow$ |
> |--------|--------|--------|--------|-----|---------|-----|-----------|
> | T2M-GPT | 0.162 | 0.281 | 0.347 | 11.193 | 6.866 | 7.751 | 2.038 |
> | InterMask | 0.449 | 0.599 | 0.683 | 5.154 | 3.790 | 7.944 | 1.737 |
>
> ---
>
> [2] MotionGPT: Human Motion as a Foreign Language, NeurIPS 2023\
> [3] ComMDM / PriorMDM: Human Motion Diffusion as a Generative Prior, ICLR 2024\
> [4] InterGen: Diffusion-based Multi-human Motion Generation under Complex Interactions, IJCV 2024\
> [5] Inter-X: Towards Versatile Human-Human Interaction Analysis, CVPR 2024\
> [6] T2M-GPT: Generating Human Motion from Textual Descriptions with Discrete Representations, CVPR 2023

---

> > ### Author Response · Authors · 2024-11-21
> > **Reply to Reviewer jS9r (3/3)**
> >
> > **Q3**: *Regarding the Inter-M Transformer structure, although there is some evidence in the ablation studies concerning the relative importance of different attention mechanisms, it may not fully clarify their specific roles and trade-offs in different interaction scenarios and data distributions.*
> >
> > **Ans**: We thank the reviewer for their insightful feedback regarding the roles and trade-offs of different attention mechanisms in the Inter-M Transformer. To complement the quantitative ablation studies provided in Section 4.2 of the paper and the qualitative ablation results on the spatio-temporal attention block provided in Appendix F, we have conducted additional qualitative ablation experiments to better understand the contribution of other attention blocks as well. These results have been added to Appendix F of the paper, and to the Section C of the supplementary video webpage.
> >
> > In these experiments, we generate three interaction scenarios—boxing, synchronized dancing, and sneaking up—while removing each attention mechanism. Our observations show that the spatio-temporal attention mechanism is crucial for handling complex poses and ensuring spatial awareness, particularly in scenarios requiring precise body coordination. Cross-attention proves vital for producing accurate reactions and maintaining realistic reaction timings relative to the partner’s actions. Lastly, self-attention appears to serve as a refining block, enhancing the overall coherence and smoothness of the generated interactions.
> >
> > We hope these additional insights and experiments clarify the specific roles of each attention mechanism and provide a more comprehensive understanding of their importance.
> >
> > ---
> >
> > *We sincerely thank you for reviewing our work. We hope that this response addresses your concerns. Please let us know if you require any further information from our side.*

---

> > > ### Comment · Reviewer_jS9r · 2024-11-26
> > >
> > > Thank you to the author for the reply, but the explanation of SMPL is insufficient. The diffusion model generates unnatural and unrealistic results, and the method proposed in this paper still has physical issues. This contradicts the claims made in the paper. I would have rated it 5.5, but since that score is not available, I will maintain my current score.

---

> > > > ### Author Response · Authors · 2024-11-28
> > > > **Thanks for the follow-up**
> > > >
> > > > We sincerely thank the reviewer for their thoughtful feedback and the opportunity to clarify further. As shown in the newly added Section A of the supplementary videos webpage, the disconnected and sudden movements do not appear in the stick figure animations, which represent the direct output from our method. These artifacts arise only after converting the joint keypoints into the SMPL format. This is not a limitation of our method and we believe it could be addressed with a better conversion algorithm that accounts for temporal consistency.
> > > >
> > > > We would also like to respectfully reiterate that our work does not claim contributions toward resolving physical plausibility issues such as penetration and sliding. We have acknowledged these limitations in the paper, and we are excited to see how future developments can build on our framework to tackle these challenges. Our primary contribution is the introduction of a novel framework for generating human interactions in discrete space, which demonstrates significant improvements in text alignment, and overall interaction quality and realism, as evidenced by both qualitative and quantitative results. If there are any claims in the paper that may appear improperly stated or unclear, we would be most grateful if the reviewers could bring them to our attention. We would be happy to correct or clarify them to ensure accuracy and transparency.
> > > >
> > > > *We deeply appreciate the reviewer’s time and insights and look forward to continuing this scientific dialogue.*

---

> ### Author Response · Authors · 2024-12-02
> **Additional Response to Reviewer jS9r**
>
> Dear Reviewer jS9r,
>
> We sincerely thank you for your valuable insights and for acknowledging the strengths of our work, including the novel 2D discrete motion token map, the use of generative masked modeling for interaction generation, and our state-of-the-art performance on benchmarks, achieved with significantly lower inference time.
>
> We also greatly appreciate you pointing out areas for improvement, and we would like to reiterate that we have made the following efforts to address your concerns:
> 1. Added more details on the two-stage masking technique used in training, including experiments demonstrating the effect of random vs. interaction masking probabilities $p_r$.
> 2. Provided additional visual results, including comparisons between our model’s output and SMPL-converted results to address the sudden movements concern, as well as 10-second-long results.
> 3. Included comparisons with the previous SOTA VQVAE-based method, T2M-GPT, demonstrating our superior performance.
> 4. Added further visual ablation study results showcasing the effects of different attention modules in the Inter-M Transformer, both in the paper and supplementary videos.
>
> We believe that that these efforts further solidify the soundness, contribution, and presentation of our paper. We hope they adequately address your concerns and we would really appreciate it if you'd consider re-evaluating our work.
>
> As a gentle reminder, *the discussion period will end in about two days, with the reviewer response period concluding in approximately 32 hours*. We would love to hear back from the reviewer before then and are happy to provide any additional clarifications if needed. Thank you again for your time and consideration.

---

### Author Response · Authors · 2024-11-22
**General Response to All Reviewers of Paper 12356**

We thank the reviewers for their valuable feedback and for highlighting both the strengths and areas of improvement in our work. We greatly appreciate the encouraging comments about our framework, particularly regarding the **novel generative masked modeling framework for human interactions** (Reviewer jS9r), the **2D motion token map** and its ability to better preserve spatio-temporal details in the discrete representation (Reviewer jS9r, LgJ2, YC4H), **state-of-the-art results on interaction generation** (Reviewer jS9r, LgJ2, YC4H), and the **lower inference time and relatively lightweight architecture** (Reviewer jS9r, LgJ2).  We have individually addressed each reviewer’s concerns, providing detailed responses and clarifications. Here, we provide a general response to summarize the key points raised and the changes made to the paper and supplementary materials as a consequence.

We would like to re-emphasize the novelty and contributions of this work:
1. **InterMask Framework**: To the best of our knowledge, we are the first to introduce generative masked modeling for human interactions in the discrete space, which provides valuable insights into this framework regarding its effectiveness, efficiency and flexibility, and sets a new direction for the field.
2. **2D discrete motion token map**: A unique 2d discrete motion representation that better captures spatio-temporal details than previous 1D token maps.
3. **Inter-M Transformer**: While not novel in architecture design, it features custom attention modules and masking techniques that are shown to be crucial for learning collaborative modeling of interactions and intra- and inter-person spatio-temporal dependencies using 2D motion tokens.
4. All the above together results in a strong, flexible and lightweight generative framework that achieves state-of-the-art results on the human interaction generation task, with a lower inference time compared to previous state-of-the-art methods. Moreover, it supports the reaction generation task seamlessly and could potentially extend to multi-person interactions with minimal modifications to only one module.

We have sincerely taken into consideration the comments and suggestions from reviewers and have incorporated the following changes in our revised manuscript.
1. Added a section on the 2D motion token map construction as Appendix J, including a detailed figure and explanation of the 2D encoder architecture and quantization process, as suggested by reviewer **LgJ2**.
2. Rectified confusion about user study participants in Section 4.1 of the main text and Appendix I in the supplementary text, which was raised by reviewer **LgJ2**.
3. Added a section on the two-stage masking technique used during training as Appendix K, with a detailed figure, explanation, and the experiment determining $p_r$, as suggested by reviewer **jS9r**.
4. Included the value of the random masking to interaction masking ratio ($p_r$) in Table 5, Appendix C.2.
5. Added more visual samples for ablations on the attention modules in Inter-M Transformer to Appendix F, to address concerns raised by reviewers  **jS9r**, **LgJ2** and **YC4H**.
6. Changed the spelling of "modelling" to "modeling" in the title for consistency with American English, as used throughout the paper.

We have also made the following additions to the supplementary video webpage.
1. Explanation of non-fluid and sudden movements in visualization results, attributed to SMPL conversion rather than the model, in Section A, as suggested by reviewers **jS9r** and **LgJ2**.
2. Included Longer generation results (10 seconds) in Section B, as suggested by reviewer **jS9r**.
3. Added ablation results of the Inter-M Transformer for each attention module in Section C, to address concerns raised by reviewers  **jS9r**, **LgJ2** and **YC4H**.
4. Results on complex and in-the-wild instructions in Section D, as suggested by reviewer **LgJ2**.

We have sincerely made our best effort to address all concerns raised by the reviewers. We hope that the additional results, insights, and clarifications strengthen our contributions and demonstrate the capability of our model as the current state-of-the-art for human interaction generation. Please do not hesitate to let us know if you have any additional questions or concerns. *If our responses have adequately addressed your concerns, we would appreciate it so much if you could re-evaluate our work, and possibly raise your score.* Thanks again for all your effort in evaluating our work.

---

### Meta-Review · Area_Chair_6qTX · 2024-12-22

**Metareview:**

The paper introduces InterMask, a novel framework for generating 3D human-human interactions from textual descriptions. It employs 2D VQ-VAE to encode motion into a 2D discrete token map and uses masked generative modeling to model spatio-temporal interdependencies. While both the masked modeling approach and transformer-based design are not invented by the authors, the use of them for human interaction modeling is creative. By carefully integrating different components, the proposed method is able to effectively capture 4D structure of humans, achieve state-of-the-art results, and maintain high efficiency. The reviewers are unanimously positive about the paper. While there are initially some concerns about the experimental evaluations and analyses, the authors did a great job during the rebuttal phase and addressed most of them. The ACs urge the authors to incorporate the feedback from the reviewers into the final version of the paper. The ACs recommend acceptance.

**Additional Comments On Reviewer Discussion:**

The reviewers originally were concerned about weak ablation studies, lack of details, etc. The authors addressed most of the concerns during the rebuttal phase.

---

### Decision · Program_Chairs · 2025-01-22

Accept (Poster)